# PROJECTION-BASED CONSTRAINED POLICY OPTIMIZATION

**Tsung-Yen Yang**
Princeton University
ty3@princeton.edu

**Justinian Rosca**
Siemens Corporation, Corporate Technology
justinian.rosca@siemens.com

**Karthik Narasimhan**
Princeton University
karthikn@princeton.edu

**Peter J. Ramadge**
Princeton University
ramadge@princeton.edu

## ABSTRACT

We consider the problem of learning control policies that optimize a reward function while satisfying constraints due to considerations of safety, fairness, or other costs. We propose a new algorithm, Projection-Based Constrained Policy Optimization (PCPO). This is an iterative method for optimizing policies in a two-step process: the first step performs a local reward improvement update, while the second step reconciles any constraint violation by projecting the policy back onto the constraint set. We theoretically analyze PCPO and provide a lower bound on reward improvement, and an upper bound on constraint violation, for each policy update. We further characterize the convergence of PCPO based on two different metrics: $L^2$ norm and Kullback-Leibler divergence. Our empirical results over several control tasks demonstrate that PCPO achieves superior performance, averaging more than 3.5 times less constraint violation and around 15% higher reward compared to state-of-the-art methods.[1]

## 1 INTRODUCTION

Recent advances in deep reinforcement learning (RL) have demonstrated excellent performance on several domains ranging from games like Go (Silver et al., 2017) and StarCraft (AlphaStar, 2019) to robotic control (Levine et al., 2016). In these settings, agents are allowed to explore the entire state space and experiment with all possible actions during training. However, in many real-world applications such as self-driving cars and unmanned aerial vehicles, considerations of safety, fairness and other costs prevent the agent from having complete freedom to explore. For instance, an autonomous car, while optimizing its driving policies, must not take any actions that could cause harm to pedestrians or property (including itself). In effect, the agent is constrained to take actions that do not violate a specified set of constraints on state-action pairs. In this work, we address the problem of learning control policies that optimize a reward function while satisfying predefined constraints.

The problem of policy learning with constraints is more challenging since directly optimizing for the reward, as in Q-Learning (Mnih et al., 2013) or policy gradient (Sutton et al., 2000), will usually violate the constraints. One approach is to incorporate constraints into the learning process by forming a constrained optimization problem. Then perform policy updates using a conditional gradient descent with line search to ensure constraint satisfaction (Achiam et al., 2017). However, the base optimization problem can become infeasible if the current policy violates the constraints. Another approach is to add a hyperparameter weighted copy of the constraints to the objective function (Tessler et al., 2018). However, this incurs the cost of extensive hyperparameter tuning.

To address the above issues, we propose projection-based constrained policy optimization (PCPO). This is an iterative algorithm that performs policy updates in two stages. The first stage maximizes reward using a trust region optimization method (*e.g.,* TRPO (Schulman et al., 2015a)) without

---

[1] For code see the project website: https://sites.google.com/view/iclr2020-pcpo

constraints. This might result in a new intermediate policy that does not satisfy the constraints. The second stage reconciles the constraint violation (if any) by projecting the policy back onto the constraint set, *i.e.,* choosing the policy in the constraint set that is closest to the selected intermediate policy. This allows efficient updates to ensure constraint satisfaction without requiring a line search (Achiam et al., 2017) or adjusting a weight (Tessler et al., 2018). Further, due to the projection step, PCPO offers efficient recovery from infeasible (*i.e.,* constraint-violating) states (e.g., due to approximation errors), which existing methods do not handle well.

We analyze PCPO theoretically and derive performance bounds for the algorithm. Specifically, based on information geometry and policy optimization theory, we construct a lower bound on reward improvement, and an upper bound on constraint violations for each policy update. We find that with a relatively small step size for each policy update, the worst-case constraint violation and reward degradation are tolerable. We further analyze two distance measures for the projection step onto the constraint set. We find that the convergence of PCPO is affected by the smallest and largest singular values of the Fisher information matrix used during training. By observing these singular values, we can choose the appropriate projection best suited to the problem.

Empirically, we compare PCPO with state-of-the-art algorithms on four different control tasks, including two Mujoco environments with safety constraints introduced by Achiam et al. (2017) and two traffic management tasks with fairness constraints introduced by Vinitsky et al. (2018). In all cases, the proposed algorithm achieves comparable or superior performance to prior approaches, averaging more reward with fewer cumulative constraint violations. For instance, across the above tasks, PCPO achieves 3.5 times fewer constraint violations and around 15% more reward. This demonstrates the ability of PCPO robustly learn constraint-satisfying policies, and represents a step towards reliable deployment of RL in real problems.

## 2 PRELIMINARIES

We frame our policy learning as a constrained Markov Decision Process (CMDP) (Altman, 1999), where policies will direct the agent to maximize the reward while minimizing the cost. We define CMDP as the tuple $< \mathcal{S}, \mathcal{A}, T, R, C >$, where $\mathcal{S}$ is the set of states, $\mathcal{A}$ is the set of actions that the agent can take, $T : \mathcal{S} \times \mathcal{A} \times \mathcal{S} \rightarrow [0, 1]$ is the transition probability of the CMDP, $R : \mathcal{S} \times \mathcal{A} \rightarrow \mathbb{R}$ is the reward function, and $C : \mathcal{S} \times \mathcal{A} \rightarrow \mathbb{R}$ is the cost function. Given the agent's current state $s$, the policy $\pi(a|s) : \mathcal{S} \rightarrow \mathcal{A}$ selects an action $a$ for the agent to take. Based on $s$ and $a$, the agent transits to the next state (denoted by $s'$) according to the state transition model $T(s'|s, a)$, and receives the reward and pays the cost, denoted by $R(s, a)$ and $C(s, a)$, respectively.

We aim to learn a policy $\pi$ that maximizes a cumulative discounted reward, denoted by

$$J^R(\pi) \doteq \mathbb{E}_{\tau \sim \pi}\big[\sum_{t=0}^{\infty} \gamma^t R(s_t, a_t)\big],$$

while satisfying constraints, *i.e.,* making a cumulative discounted cost constraint below a desired threshold $h$, denoted by

$$J^C(\pi) \doteq \mathbb{E}_{\tau \sim \pi}\big[\sum_{t=0}^{\infty} \gamma^t C(s_t, a_t)\big] \leq h,$$

where $\gamma$ is the discount factor, $\tau$ is the trajectory ($\tau = (s_0, a_0, s_1, \cdots)$), and $\tau \sim \pi$ is shorthand for showing that the distribution over the trajectory depends on $\pi : s_0 \sim \mu, a_t \sim \pi(a_t|s_t), s_{t+1} \sim T(s_{t+1}|s_t, a_t)$, where $\mu$ is the initial state distribution.

Kakade & Langford (2002) give an identity to express the performance of policy $\pi'$ in terms of the advantage function over another policy $\pi$ :

$$J^R(\pi') - J^R(\pi) = \frac{1}{1-\gamma}\mathbb{E}_{\substack{s \sim d^{\pi'} \\ a \sim \pi'}}[A_R^{\pi}(s, a)], \tag{1}$$

where $d^{\pi}$ is the discounted future state distribution, denoted by $d^{\pi}(s) \doteq (1 - \gamma)\sum_{t=0}^{\infty}\gamma^t P(s_t = s|\pi)$, and $A_R^{\pi}(s, a)$ is the reward advantage function, denoted by $A_R^{\pi}(s, a) \doteq Q_R^{\pi}(s, a) - V_R^{\pi}(s)$. Here $Q_R^{\pi}(s, a) \doteq \mathbb{E}_{\tau \sim \pi}\big[\sum_{t=0}^{\infty}\gamma^t R(s_t, a_t)|s_0 = s, a_0 = a\big]$ is the discounted cumulative reward obtained by the policy $\pi$ given the initial state $s$ and action $a$, and $V_R^{\pi}(s) \doteq$

$\mathbb{E}_{\tau \sim \pi}\left[\sum_{t=0}^{\infty} \gamma^t R(s_t, a_t)|s_0 = s\right]$ is the discounted cumulative reward obtained by the policy $\pi$ given the initial state $s$. Similarly, we have the cost advantage function $A_C^\pi(s, a) = Q_C^\pi(s, a) - V_C^\pi(s)$, where $Q_C^\pi(s, a) \doteq \mathbb{E}_{\tau \sim \pi}\left[\sum_{t=0}^{\infty} \gamma^t C(s_t, a_t)|s_0 = s, a_0 = a\right]$, and $V_C^\pi(s) \doteq \mathbb{E}_{\tau \sim \pi}\left[\sum_{t=0}^{\infty} \gamma^t C(s_t, a_t)|s_0 = s\right]$.

## 3 PROJECTION-BASED CONSTRAINED POLICY OPTIMIZATION

To robustly learn constraint-satisfying policies, we develop PCPO – a trust region method that performs policy updates corresponding to reward improvement, followed by projections onto the constraint set. PCPO, inspired by *projected gradient descent*, is composed of two steps for each update, a reward improvement step and a projection step (This is illustrated in Fig. 1).

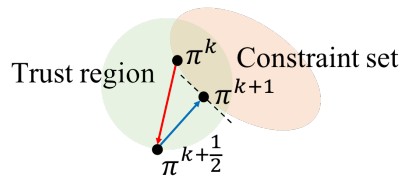

Figure 1: Update procedures for PCPO. In step one (red arrow), PCPO follows the reward improvement direction in the trust region (light green). In step two (blue arrow), PCPO projects the policy onto the constraint set (light orange).

**Reward Improvement Step.** First, we optimize the reward function by maximizing the reward advantage function $A_R^\pi(s, a)$ subject to a Kullback-Leibler (KL) divergence constraint. This constraints the intermediate policy $\pi^{k+\frac{1}{2}}$ to be within a $\delta$-neighbourhood of $\pi^k$:

$$\pi^{k+\frac{1}{2}} = \arg\max_\pi \quad \mathbb{E}_{\substack{s \sim d^{\pi^k} \\ a \sim \pi}}[A_R^{\pi^k}(s, a)]$$
$$\text{s.t.} \quad \mathbb{E}_{s \sim d^{\pi^k}}\left[D_{\mathrm{KL}}(\pi||\pi^k)[s]\right] \leq \delta. \tag{2}$$

This update rule with the trust region, $\{\pi : \mathbb{E}_{s \sim d^{\pi^k}}\left[D_{\mathrm{KL}}(\pi||\pi^k)[s]\right] \leq \delta\}$, is called Trust Region Policy Optimization (TRPO) (Schulman et al., 2015a). It constraints the policy changes to a divergence neighborhood and guarantees reward improvement.

**Projection Step.** Second, we project the intermediate policy $\pi^{k+\frac{1}{2}}$ onto the constraint set by minimizing a distance measure $D$ between $\pi^{k+\frac{1}{2}}$ and $\pi$:

$$\pi^{k+1} = \arg\min_\pi \quad D(\pi, \pi^{k+\frac{1}{2}})$$
$$\text{s.t.} \quad J^C(\pi^k) + \mathbb{E}_{\substack{s \sim d^{\pi^k} \\ a \sim \pi}}[A_C^{\pi^k}(s, a)] \leq h. \tag{3}$$

The projection step ensures that the constraint-satisfying policy $\pi^{k+1}$ is close to $\pi^{k+\frac{1}{2}}$. We consider two distance measures $D$: $L^2$ norm and KL divergence. In contrast, using KL divergence projection in the probability distribution space allows us to provide provable guarantees for PCPO.

### 3.1 PERFORMANCE BOUND FOR PCPO WITH KL DIVERGENCE PROJECTION

In safety-critical applications such as autonomous cars, one cares about how worse the performance of a system evolves when applying a learning algorithm. To this end, for PCPO with KL divergence projection, we analyze the worst-case performance degradation for each policy update when the current policy $\pi^k$ *satisfies* the constraint. The following theorem provides a lower bound on reward improvement, and an upper bound on constraint violation for each policy update.

**Theorem 3.1** (**Worst-case Bound on Updating Constraint-satisfying Policies**). *Define $\epsilon_R^{\pi^{k+1}} \doteq \max_s \left|\mathbb{E}_{a \sim \pi^{k+1}}[A_R^{\pi^k}(s, a)]\right|$, and $\epsilon_C^{\pi^{k+1}} \doteq \max_s \left|\mathbb{E}_{a \sim \pi^{k+1}}[A_C^{\pi^k}(s, a)]\right|$. If the current policy $\pi^k$ satisfies the constraint, then under KL divergence projection, the lower bound on reward improvement, and upper bound on constraint violation for each policy update are*

$$J^R(\pi^{k+1}) - J^R(\pi^k) \geq -\frac{\sqrt{2\delta}\gamma\epsilon_R^{\pi^{k+1}}}{(1-\gamma)^2}, \text{ and } J^C(\pi^{k+1}) \leq h + \frac{\sqrt{2\delta}\gamma\epsilon_C^{\pi^{k+1}}}{(1-\gamma)^2},$$

*where $\delta$ is the step size in the reward improvement step.*

*Proof.* See the supplemental material. □

Theorem 3.1 indicates that if $\delta$ is small, the worst-case performance degradation is tolerable.

Due to approximation errors or the random initialization of policies, PCPO may have a constraint-violating update. Theorem 3.1 does not give the guarantee on updating a *constraint-violating* policy. Hence we analyze worst-case performance degradation for each policy update when the current policy $\pi^k$ *violates* the constraint. The following theorem provides a lower bound on reward improvement, and an upper bound on constraint violation for each policy update.

**Theorem 3.2** (**Worst-case Bound on Updating Constraint-violating Policies**). *Define* $\epsilon_R^{\pi^{k+1}} \doteq \max_s \left| \mathbb{E}_{a \sim \pi^{k+1}} [A_R^{\pi^k}(s,a)] \right|$, $\epsilon_C^{\pi^{k+1}} \doteq \max_s \left| \mathbb{E}_{a \sim \pi^{k+1}} [A_C^{\pi^k}(s,a)] \right|$, $b^+ \doteq \max(0, J^C(\pi^k) - h)$, *and* $\alpha_{\mathrm{KL}} \doteq \frac{1}{2a^T H^{-1} a}$, *where* $a$ *is the gradient of the cost advantage function and* $H$ *is the Hessian of the KL divergence constraint. If the current policy* $\pi^k$ *violates the constraint, then under KL divergence projection, the lower bound on reward improvement and the upper bound on constraint violation for each policy update are*

$$J^R(\pi^{k+1}) - J^R(\pi^k) \geq - \frac{\sqrt{2(\delta + b^{+^2}\alpha_{\mathrm{KL}})}\gamma\epsilon_R^{\pi^{k+1}}}{(1-\gamma)^2},$$

$$and \ J^C(\pi^{k+1}) \leq h + \frac{\sqrt{2(\delta + b^{+^2}\alpha_{\mathrm{KL}})}\gamma\epsilon_C^{\pi^{k+1}}}{(1-\gamma)^2},$$

*where* $\delta$ *is the step size in the reward improvement step.*

*Proof.* See the supplemental material. □

Theorem 3.2 indicates that when the policy has greater constraint violation ($b^+$ increases), its worst-case performance degradation increases. Note that Theorem 3.2 reduces to Theorem 3.1 if the current policy $\pi^k$ satisfies the constraint ($b^+ = 0$). The proofs of Theorem 3.1 and Theorem 3.2 follow from the fact that the projection of the policy is non-expansive, *i.e.,* the distance between the projected policies is smaller than that of the unprojected policies. This allows us to measure it and bound the KL divergence between the current policy and the new policy.

## 4 PCPO Updates

For a large neural network policy with many parameters, it is impractical to directly solve for the PCPO update in Problem 2 and Problem 3 due to the computational cost. However, with a small step size $\delta$, we can approximate the reward function and constraints with a first order expansion, and approximate the KL divergence constraint in the reward improvement step, and the KL divergence measure in the projection step with a second order expansion. We now make several definitions:

$g \doteq \nabla_\theta \mathbb{E}_{\substack{s \sim d^{\pi^k} \\ a \sim \pi}} [A_R^{\pi^k}(s,a)]$ is the gradient of the reward advantage function,

$a \doteq \nabla_\theta \mathbb{E}_{\substack{s \sim d^{\pi^k} \\ a \sim \pi}} [A_C^{\pi^k}(s,a)]$ is the gradient of the cost advantage function,

$H_{i,j} \doteq \frac{\partial^2 \mathbb{E}_{s \sim d^{\pi^k}} \left[ D_{\mathrm{KL}}(\pi||\pi^k)[s] \right]}{\partial \theta_j \partial \theta_j}$ is the Hessian of the KL divergence constraint ($H$ is also called the Fisher information matrix. It is symmetric positive semi-definite), $b \doteq J^C(\pi^k) - h$ is the constraint violation of the policy $\pi^k$, and $\theta$ is the parameter of the policy.

**Reward Improvement Step.** We linearize the objective function at $\pi^k$ subject to second order approximation of the KL divergence constraint in order to obtain the following updates:

$$\theta^{k+\frac{1}{2}} = \arg\max_\theta \ \ g^T(\theta - \theta^k)$$

$$\text{s.t.} \quad \frac{1}{2}(\theta - \theta^k)^T H(\theta - \theta^k) \leq \delta. \tag{4}$$

---

**Algorithm 1** Projection-Based Constrained Policy Optimization (PCPO)

> Initialize policy $\pi^0 = \pi(\boldsymbol{\theta}^0)$
> **for** $k = 0, 1, 2, \cdots$ **do**
>     Run $\pi^k = \pi(\boldsymbol{\theta}^k)$ and store trajectories in $\mathcal{D}$
>     Compute $\boldsymbol{g}, \boldsymbol{a}, \boldsymbol{H}$, and $b$ using $\mathcal{D}$
>     Obtain $\boldsymbol{\theta}^{k+1}$ using update in Eq. (6)
>     Empty $\mathcal{D}$

---

**Projection Step.** If the projection is defined in the parameter space, we can directly use $L^2$ norm projection. On the other hand, if the projection is defined in the probability space, we can use KL divergence projection. This can be approximated through the second order expansion. Again, we linearize the cost constraint at $\pi^k$. This gives the following update for the projection step:

$$\boldsymbol{\theta}^{k+1} = \arg\min_{\boldsymbol{\theta}} \quad \frac{1}{2}(\boldsymbol{\theta} - \boldsymbol{\theta}^{k+\frac{1}{2}})^T \boldsymbol{L}(\boldsymbol{\theta} - \boldsymbol{\theta}^{k+\frac{1}{2}})$$

$$\text{s.t.} \quad \boldsymbol{a}^T(\boldsymbol{\theta} - \boldsymbol{\theta}^k) + b \leq 0, \tag{5}$$

where $\boldsymbol{L} = \boldsymbol{I}$ for $L^2$ norm projection, and $\boldsymbol{L} = \boldsymbol{H}$ for KL divergence projection. One may argue that using linear approximation to the constraint set is not enough to ensure constraint satisfaction since the real constraint set is maybe non-convex. However, if the step size $\delta$ is small, then the linearization of the constraint set is accurate enough to locally approximate it.

We solve Problem (4) and Problem (5) using convex programming (See the supplemental material for the derivation). For each policy update, we have

$$\boldsymbol{\theta}^{k+1} = \boldsymbol{\theta}^k + \sqrt{\frac{2\delta}{\boldsymbol{g}^T \boldsymbol{H}^{-1} \boldsymbol{g}}} \boldsymbol{H}^{-1} \boldsymbol{g} - \max\left(0, \frac{\sqrt{\frac{2\delta}{\boldsymbol{g}^T \boldsymbol{H}^{-1} \boldsymbol{g}}} \boldsymbol{a}^T \boldsymbol{H}^{-1} \boldsymbol{g} + b}{\boldsymbol{a}^T \boldsymbol{L}^{-1} \boldsymbol{a}}\right) \boldsymbol{L}^{-1} \boldsymbol{a}. \tag{6}$$

We assume that $\boldsymbol{H}$ does not have $0$ as an eigenvalue and hence it is invertible. PCPO requires to invert $\boldsymbol{H}$, which is impractical for huge neural network policies. Hence we use the conjugate gradient method (Schulman et al., 2015a). Algorithm 1 shows the pseudocode. (See supplemental material for a discussion of the tradeoff between the approximation error and computational efficiency of the conjugate gradient method.)

**Analysis of PCPO Update Rule.** For a problem including multiple constraints, we can extend the update in Eq. (6) by using alternating projections. This approach finds a solution in the intersection of multiple constraint sets by sequentially projecting onto each of the sets. The update rule in Eq. (6) shows that the difference between PCPO with KL divergence and $L^2$ norm projections is the cost update direction, leading to a difference in reward improvement. These two projections converge to different stationary points with different convergence rates related to the smallest and largest singular values of the Fisher information matrix shown in Theorem 4.1. For our analysis, we make the following assumptions: we *minimize* the negative reward objective function $f : \mathbb{R}^n \to \mathbb{R}$ (We follow the convention of the literature that authors typically minimize the objective function). The function $f$ is $L$-smooth and twice continuously differentiable over the closed and convex constraint set $\mathcal{C}$.

**Theorem 4.1 (Reward Improvement Under $L^2$ Norm and KL Divergence Projections).** *Let $\eta \doteq \sqrt{\frac{2\delta}{\boldsymbol{g}^T \boldsymbol{H}^{-1} \boldsymbol{g}}}$ in Eq. (6), where $\delta$ is the step size for reward improvement, $\boldsymbol{g}$ is the gradient of $f$, and $\boldsymbol{H}$ is the Fisher information matrix. Let $\sigma_{\max}(\boldsymbol{H})$ be the largest singular value of $\boldsymbol{H}$, and $\boldsymbol{a}$ be the gradient of cost advantage function in Eq. (6). Then PCPO with KL divergence projection converges to a stationary point either inside the constraint set or in the boundary of the constraint set. In the latter case, the Lagrangian constraint $\boldsymbol{g} = -\alpha\boldsymbol{a}, \alpha \geq 0$ holds. Moreover, at step $k + 1$ the objective value satisfies*

$$f(\boldsymbol{\theta}^{k+1}) \leq f(\boldsymbol{\theta}^k) + ||\boldsymbol{\theta}^{k+1} - \boldsymbol{\theta}^k||^2_{-\frac{1}{\eta}\boldsymbol{H} + \frac{L}{2}\boldsymbol{I}}.$$

*PCPO with $L^2$ norm projection converges to a stationary point either inside the constraint set or in the boundary of the constraint set. In the latter case, the Lagrangian constraint $\boldsymbol{H}^{-1}\boldsymbol{g} = -\alpha\boldsymbol{a}, \alpha \geq$*

0 *holds. If* $\sigma_{\max}(\boldsymbol{H}) \leq 1$, *then a step* $k+1$ *objective value satisfies*

$$f(\boldsymbol{\theta}^{k+1}) \leq f(\boldsymbol{\theta}^k) + (\frac{L}{2} - \frac{1}{\eta})||\boldsymbol{\theta}^{k+1} - \boldsymbol{\theta}^k||_2^2.$$

*Proof.* See the supplemental material. □

Theorem 4.1 shows that in the stationary point $\boldsymbol{g}$ is a line that points to the opposite direction of $\boldsymbol{a}$. Further, the improvement of the objective value is affected by the singular value of the Fisher information matrix. Specifically, the objective of KL divergence projection decreases when $\frac{L\eta}{2}\boldsymbol{I} \prec \boldsymbol{H}$, implying that $\sigma_{\min}(\boldsymbol{H}) > \frac{L\eta}{2}$. And the objective of $L^2$ norm projection decreases when $\eta < \frac{2}{L}$, implying that condition number of $\boldsymbol{H}$ is upper bounded: $\frac{\sigma_{\max}(\boldsymbol{H})}{\sigma_{\min}(\boldsymbol{H})} < \frac{2||\boldsymbol{g}||_2^2}{L^2\delta}$. Observing the singular values of the Fisher information matrix allows us to adaptively choose the appropriate projection and hence achieve objective improvement. In the supplemental material, we further use an example to compare the optimization trajectories and stationary points of KL divergence and $L^2$ norm projections.

## 5 RELATED WORK

**Policy Learning with Constraints.** Learning constraint-satisfying policies has been explored in the context of safe RL (Garcia & Fernandez, 2015). The agent learns policies either by (1) exploration of the environment (Achiam et al., 2017; Tessler et al., 2018; Chow et al., 2017) or (2) through expert demonstrations (Ross et al., 2011; Rajeswaran et al., 2017; Gao et al., 2018). However, using expert demonstrations requires humans to label the constraint-satisfying behavior for every possible situation. The scalability of these rule-based approaches is an issue since many real autonomous systems such as self-driving cars and industrial robots are inherently complex. To overcome this issue, PCPO uses the first approach in which the agent learns by trial and error. To prevent the agent from having constraint-violating behavior during exploring the environment, PCPO uses the projection onto the constraint set to ensure constraint satisfaction throughout learning.

**Constraint satisfaction by Projections.** Using a projection onto a constraint set has been explored for general constrained optimization in other contexts. For example, Akrour et al. (2019) projects the policy from a parameter space onto the constraint. This ensures the updated policy stays close to the previous policy. In contrast, we examine constraints that are defined in terms of states and actions. Similarly, Chow et al. (2019) proposes $\theta$-projection. This approach projects the policy parameters $\theta$ onto the constraint set. However, no provide provable guarantees are provided. Moreover, the

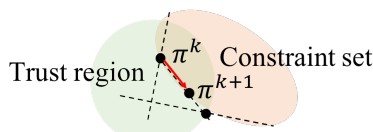

Figure 2: Update procedures for CPO (Achiam et al., 2017). CPO computes the update by simultaneously considering the trust region (light green) and the constraint set (light orange). CPO becomes infeasible when these two sets do not intersect.

problem is formulated by adding the weighted constraint to the reward objective function. Since the weight must be tuned, this incurs the cost of hyperparameter tuning. In contrast, PCPO eliminates the cost of the hyperparameter tuning, and provides provable guarantees on learning constraint-satisfying policies.

**Comparison to CPO (Achiam et al., 2017).** Perhaps the closest work to ours is the approach of Achiam et al. (2017), who proposes the constrained policy optimization (CPO) algorithm to solve the following:

$$\boldsymbol{\theta}^{k+1} = \arg\max_{\boldsymbol{\theta}} \boldsymbol{g}^T(\boldsymbol{\theta} - \boldsymbol{\theta}^k) \quad \text{s.t.} \frac{1}{2}(\boldsymbol{\theta} - \boldsymbol{\theta}^k)^T \boldsymbol{H}(\boldsymbol{\theta} - \boldsymbol{\theta}^k) \leq \delta, \ \boldsymbol{a}^T(\boldsymbol{\theta} - \boldsymbol{\theta}^k) + b \leq 0. \quad (7)$$

CPO simultaneously considers the trust region and the constraint, and uses the line search to select a step size (This is illustrated in Fig. 2). The update rule of CPO becomes infeasible when the current policy violates the constraint ($b > 0$). CPO recovers by replacing Problem (7) with an update to purely decrease the constraint value: $\boldsymbol{\theta}^{k+1} = \boldsymbol{\theta}^k - \sqrt{\frac{2\delta}{\boldsymbol{a}^T\boldsymbol{H}^{-1}\boldsymbol{a}}}\boldsymbol{H}^{-1}\boldsymbol{a}$. This update rule may lead

(a) Gather      (b) Circle      (c) Grid      (d) Bottleneck

Figure 3: The gather, circle, grid and bottleneck tasks. (a) Gather task: the agent is rewarded for gathering green apples but is constrained to collect a limited number of red fruit (Achiam et al., 2017). (b) Circle task: the agent is rewarded for moving in a specified wide circle, but is constrained to stay within a safe region smaller than the radius of the circle (Achiam et al., 2017). (c) Grid task: the agent controls the traffic lights in a grid road network and is rewarded for high throughput but constrained to let lights stay red for at most 7 consecutive seconds (Vinitsky et al., 2018). (d) Bottleneck task: the agent controls a set of autonomous vehicles (shown in red) in a traffic merge situation and is rewarded for achieving high throughput but constrained to ensure that human-driven vehicles (shown in white) have low speed for no more than 10 seconds (Vinitsky et al., 2018).

to a slow progress in learning constraint-satisfying policies. In contrast, PCPO first optimizes the reward and uses the projection to satisfy the constraint. This ensures a feasible solution, allowing the agent to improve the reward while ensuring constraint satisfaction simultaneously.

## 6 EXPERIMENTS

**Tasks.** We compare the proposed algorithm with existing approaches on four control tasks in total: two tasks with safety constraints ((a) and (b) in Fig. 3), and two tasks with fairness constraints ((c) and (d) in Fig. 3). These tasks are briefly described in the caption of Fig. 3. The first two tasks – *Gather* and *Circle* – are Mujoco environments with state space constraints introduced by Achiam et al. (2017). The other two tasks – *Grid* and *Bottleneck* – are traffic management problems where the agent controls either a traffic light or a fleet of autonomous vehicles. This is especially challenging since the dimensions of state and action spaces are larger, and the dynamics of the environment are inherently complex.

**Baselines.** We compare PCPO with four baselines outlined below.

(1) Constrained Policy Optimization (CPO) (Achiam et al., 2017).

(2) Primal-dual Optimization (PDO) (Chow et al., 2017). In PDO, the weight (dual variables) is learned based on the current constraint satisfaction. A PDO policy update solves:

$$\boldsymbol{\theta}^{k+1} = \arg\max_{\boldsymbol{\theta}} \quad \boldsymbol{g}^T(\boldsymbol{\theta} - \boldsymbol{\theta}^k) + \lambda^k \boldsymbol{a}^T(\boldsymbol{\theta} - \boldsymbol{\theta}^k), \tag{8}$$

where $\lambda^k$ is updated using $\lambda^{k+1} = \lambda^k + \beta(J^C(\pi^k) - h)$. Here $\beta$ is a fixed learning rate.

(3) Fixed-point Policy Optimization (FPO). A variant of PDO that solves Eq. (8) using a constant $\lambda$.

(4) Trust Region Policy Optimization (TRPO) (Schulman et al., 2015a). The TRPO policy update is an *unconstrained* one:

$$\boldsymbol{\theta}^{k+1} = \boldsymbol{\theta}^k + \sqrt{\tfrac{2\delta}{\boldsymbol{g}^T \boldsymbol{H}^{-1} \boldsymbol{g}}} \boldsymbol{H}^{-1}\boldsymbol{g}.$$

Note that TRPO ignores any constraints. We include it to serve as an upper bound baseline on the reward performance.

Since the main focus is to compare PCPO with the state-of-the-art algorithm, CPO, PDO and FPO are not shown in the ant circle, ant gather, grid and bottleneck tasks for clarity.

**Experimental Details.** For the gather and circle tasks we test two distinct agents: a point-mass ($S \subseteq \mathbb{R}^9, A \subseteq \mathbb{R}^2$), and an ant robot ($S \subseteq \mathbb{R}^{32}, A \subseteq \mathbb{R}^8$). The agent in the grid task is $S \subseteq \mathbb{R}^{156}, A \subseteq \mathbb{R}^4$, and the agent in bottleneck task is $S \subseteq \mathbb{R}^{141}, A \subseteq \mathbb{R}^{20}$. For the simulations in the gather and circle tasks, we use a neural network with two hidden layers of size (64, 32) to represent

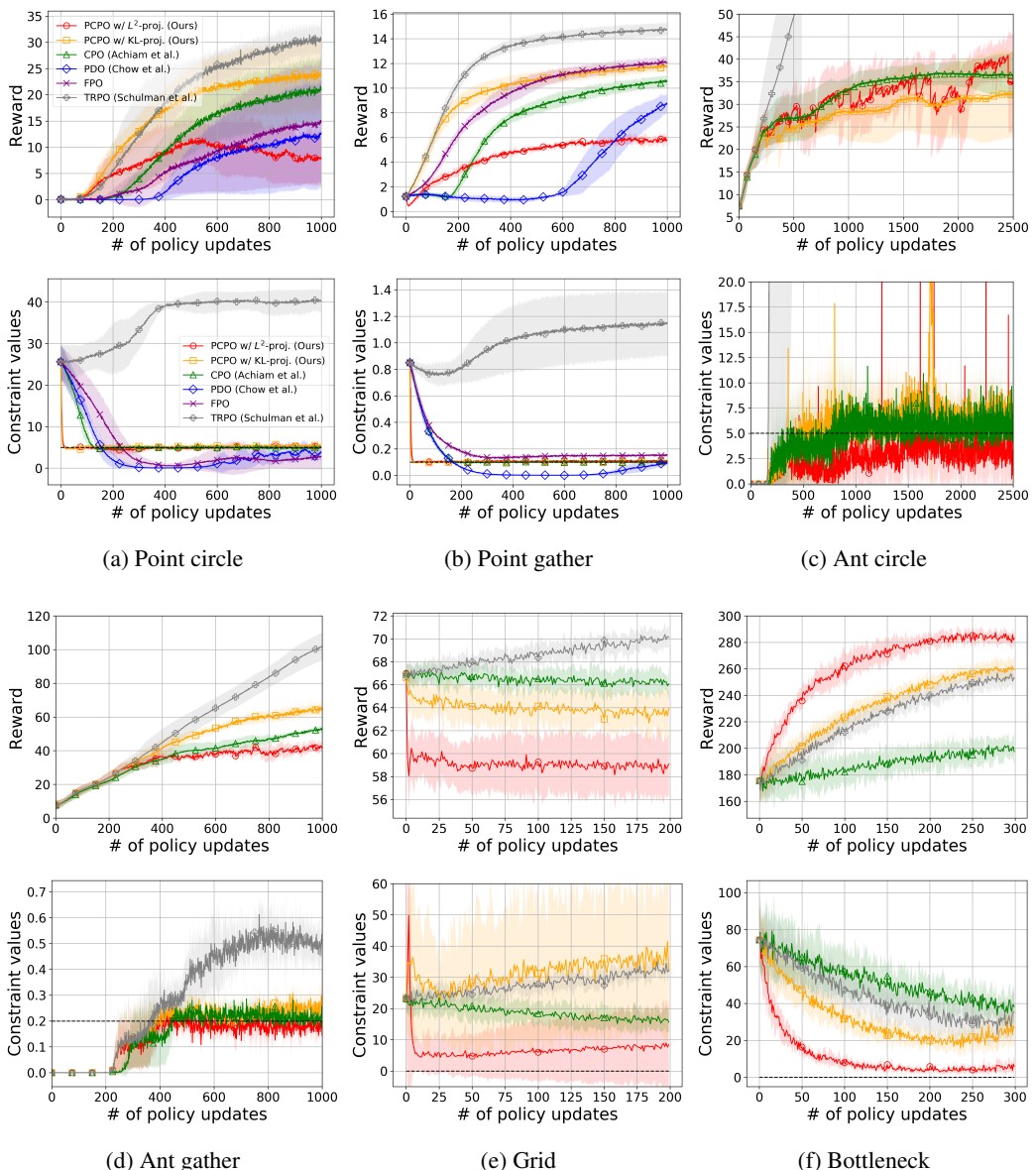

Figure 4: The values of the discounted reward and the undiscounted constraint value (the total number of constraint violation) along policy updates for the tested algorithms and task pairs. The solid line is the mean and the shaded area is the standard deviation, over five runs. The dashed line in the cost constraint plot is the cost constraint threshold $h$. The curves for baseline oracle, TRPO, indicate the reward and constraint violation values when the constraint is *ignored*. (Best viewed in color, and the legend is shared across all the figures.)

Gaussian policies. For the simulations in the grid and bottleneck tasks, we use a neural network with two hidden layers of size (16, 16) and (50,25) to represent Gaussian policies, respectively. In the experiments, since the step size is small, we reuse the Fisher information matrix of the reward improvement step in the KL projection step to reduce the computational cost. The step size $\delta$ is set to $10^{-4}$ for all tasks and all tested algorithms. For each task, we conduct 5 runs to get the mean and standard deviation for both the reward and the constraint value over the policy updates. The experiments are implemented in rllab (Duan et al., 2016), a tool for developing and evaluating RL algorithms. See the supplemental material for the details of the experiments.

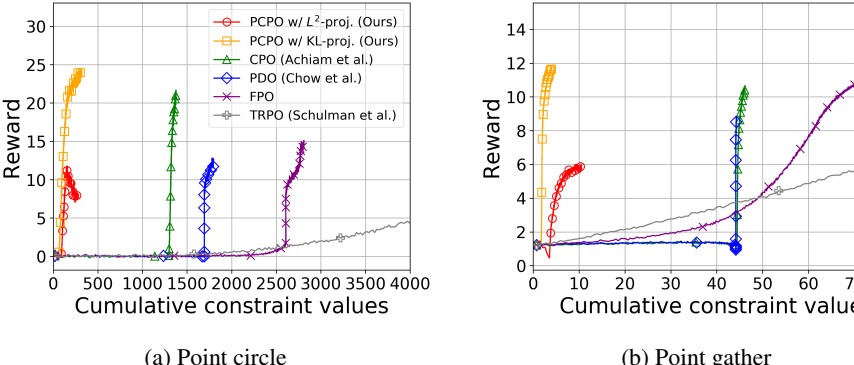

(a) Point circle          (b) Point gather

Figure 5: The value of the discounted reward versus the cumulative constraint value for the tested algorithms and task pairs. See the supplemental material for learning curves in the other tasks. PCPO achieves less constraint violation under the same reward improvement compared to the other algorithms.

**Overall Performance.** The learning curves of the discounted reward and the undiscounted constraint value (the total number of constraint violation) over policy updates are shown for all tested algorithms and tasks in Fig. 4. The dashed line in the constraint figure is the cost constraint threshold $h$. The curves for baseline oracle, TRPO, indicate the reward and constraint value when the constraint is ignored. Overall, we find that PCPO is able to improve the reward while having the fastest constraint satisfaction in all tasks. In particular, PCPO is the only algorithm that learns constraint-satisfying policies across all the tasks. Moreover we observe that (1) CPO has more constraint violation than PCPO, (2) PDO is too conservative in optimizing the reward, and (3) FPO requires a significant effort to select a good value of $\lambda$.

We also observe that in Grid and Bottleneck task, there is slightly more constraint violation than the easier task such as point circle and point gather. This is due to complexity of the policy behavior and non-convexity of the constraint set. However, even with a linear approximation of the constraint set, PCPO still outperforms CPO with 85.15% and 5.42 times less constraint violation in Grid and Bottleneck task, respectively.

These observations suggest that projection step in PCPO drives the agent to learn the constraint-satisfying policy within few policy updates, giving PCPO an advantage in applications. To show that PCPO achieves the same reward with less constraint violation, we examine the reward versus the cumulative constraint value for the tested algorithms in point circle and point gather task shown in Fig. 5. We observe that PCPO outperforms CPO significantly with 66 times and 15 times less constraint violation under the same reward improvement in point circle and point gather tasks, respectively. This observation suggests that PCPO enables the agent to cautiously explore the environment under the constraints.

**Comparison of PCPO with KL Divergence vs. $L^2$ Norm Projections.** We observe that PCPO with $L^2$ norm projection is more constraint-satisfying than PCPO with KL divergence projection. In addition, PCPO with $L^2$ norm projection tends to have reward fluctuation (point circle, ant circle, and ant gather tasks), while with KL divergence projection tends to have more stable reward improvement (all the tasks).

The above observations indicate that since the gradient of constraint is not multiplied by the Fisher information matrix, the gradient of the constraint is not aligned with the gradient of the reward. This reduces the reward improvement. However, when the Fisher information matrix is ill-conditioned or not well-estimated, especially in a high dimensional policy space, a bad constraint update direction may hinder constraint satisfaction (ant circle, ant gather, grid and bottleneck tasks). In addition, since the stationary points of KL divergence and $L^2$ norm projections are different, they converge to policies with different reward (observe that PCPO with $L^2$ norm projection has higher reward than the one with KL divergence projection around 2250 iterations in ant circle task, and has less reward in point gather task).

**Discussion of PDO and FPO.** For the PDO baseline, we see that its constraint values fluctuate especially in the point circle task. This phenomena suggests that PDO is not able to adjust the weight $\lambda^k$ quickly enough to meet the constraint threshold, which hinders the efficiency of learning constraint-satisfying policies. If the learning rate $\beta$ is too big, the agent will be too conservative in improving the reward. For FPO, we also see that it learns near constraint-satisfying policies with slightly larger reward improvement compared to PDO. However, in practice FPO requires a lot of engineering effort to select a good value of $\lambda$. Since PCPO requires no hyperparameter tuning, it has the advantage of robustly learning constraint-satisfying policies over PDO and FPO.

## 7 CONCLUSION

We address the problem of finding constraint-satisfying policies. The proposed algorithm – projection-based constrained policy optimization (PCPO) – optimizes for the reward function while using the projections to ensure constraint satisfaction. This update rule allows PCPO to maintain the feasibility of the optimization problem of each update, addressing the issue of state-of-the-art approaches. The algorithm achieves comparable or superior performance to state-of-the-art approaches in terms of reward improvement and constraint satisfaction in all cases. We further analyze the convergence of PCPO, and find that certain tasks may prefer either KL divergence projection or $L^2$ norm projection. Future work will consider the following: (1) examining the Fisher information matrix to iteratively prescribe the choice of projection for policy update, and hence robustly learn constraint-satisfying policies with more reward improvement, and (2) using expert demonstration or other domain knowledge to reduce the sample complexity.

### ACKNOWLEDGMENTS

The authors would like to thank the anonymous reviewers and the area chair for their comments. Tsung-Yen Yang thanks Siemens Corporation, Corporate Technology for their support.

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

# S Supplementary Materials

## S.1 Proof of Theorem 3.1: Performance Bound on Updating the Constraint-satisfying Policy

To prove the policy performance bound when the current policy is feasible (*i.e.,* constraint-satisfying), we prove the KL divergence between $\pi^k$ and $\pi^{k+1}$ for the KL divergence projection. We then prove our main theorem for the worst-case performance degradation.

**Lemma S.1.** *If the current policy $\pi^k$ satisfies the constraint, the constraint set is closed and convex, the KL divergence constraint for the first step is $\mathbb{E}_{s \sim d^{\pi^k}}\left[D_{\mathrm{KL}}(\pi^{k+\frac{1}{2}}||\pi^k)[s]\right] \leq \delta$, where $\delta$ is the step size in the reward improvement step, then under KL divergence projection, we have*

$$\mathbb{E}_{s \sim d^{\pi^k}}\left[D_{\mathrm{KL}}(\pi^{k+1}||\pi^k)[s]\right] \leq \delta.$$

*Proof.* By the Bregman divergence projection inequality, $\pi^k$ being in the constraint set, and $\pi^{k+1}$ being the projection of the $\pi^{k+\frac{1}{2}}$ onto the constraint set, we have

$$\mathbb{E}_{s \sim d^{\pi^k}}\left[D_{\mathrm{KL}}(\pi^k||\pi^{k+\frac{1}{2}})[s]\right] \geq \mathbb{E}_{s \sim d^{\pi^k}}\left[D_{\mathrm{KL}}(\pi^k||\pi^{k+1})[s]\right] + \mathbb{E}_{s \sim d^{\pi^k}}\left[D_{\mathrm{KL}}(\pi^{k+1}||\pi^{k+\frac{1}{2}})[s]\right]$$

$$\Rightarrow \delta \geq \mathbb{E}_{s \sim d^{\pi^k}}\left[D_{\mathrm{KL}}(\pi^k||\pi^{k+\frac{1}{2}})[s]\right] \geq \mathbb{E}_{s \sim d^{\pi^k}}\left[D_{\mathrm{KL}}(\pi^k||\pi^{k+1})[s]\right].$$

The derivation uses the fact that KL divergence is always greater than zero. We know that KL divergence is asymptotically symmetric when updating the policy within a local neighbourhood. Thus, we have

$$\delta \geq \mathbb{E}_{s \sim d^{\pi^k}}\left[D_{\mathrm{KL}}(\pi^{k+\frac{1}{2}}||\pi^k)[s]\right] \geq \mathbb{E}_{s \sim d^{\pi^k}}\left[D_{\mathrm{KL}}(\pi^{k+1}||\pi^k)[s]\right].$$

$\square$

Now we use Lemma S.1 to prove our main theorem.

**Theorem S.2.** *Define $\epsilon_R^{\pi^{k+1}} \doteq \max_s \left|\mathbb{E}_{a \sim \pi^{k+1}}[A_R^{\pi^k}(s,a)]\right|$, and $\epsilon_C^{\pi^{k+1}} \doteq \max_s \left|\mathbb{E}_{a \sim \pi^{k+1}}[A_C^{\pi^k}(s,a)]\right|$. If the current policy $\pi^k$ satisfies the constraint, then under the KL divergence projection, the lower bound on reward improvement, and upper bound on constraint violation for each policy update are*

$$J^R(\pi^{k+1}) - J^R(\pi^k) \geq -\frac{\sqrt{2\delta}\gamma\epsilon_R^{\pi^{k+1}}}{(1-\gamma)^2}, \text{ and } J^C(\pi^{k+1}) \leq h + \frac{\sqrt{2\delta}\gamma\epsilon_C^{\pi^{k+1}}}{(1-\gamma)^2},$$

*where $\delta$ is the step size in the reward improvement step.*

*Proof.* By the theorem in Achiam et al. (2017) and Lemma S.1, we have the following reward degradation bound for each policy update:

$$J^R(\pi^{k+1}) - J^R(\pi^k) \geq \frac{1}{1-\gamma}\mathbb{E}_{\substack{s \sim d^{\pi^k} \\ a \sim \pi^{k+1}}}\left[A_R^{\pi^k}(s,a) - \frac{2\gamma\epsilon_R^{\pi^{k+1}}}{1-\gamma}\sqrt{\frac{1}{2}D_{\mathrm{KL}}(\pi^{k+1}||\pi^k)[s]}\right]$$

$$\geq \frac{1}{1-\gamma}\mathbb{E}_{\substack{s \sim d^{\pi^k} \\ a \sim \pi^{k+1}}}\left[-\frac{2\gamma\epsilon_R^{\pi^{k+1}}}{1-\gamma}\sqrt{\frac{1}{2}D_{\mathrm{KL}}(\pi^{k+1}||\pi^k)[s]}\right]$$

$$\geq -\frac{\sqrt{2\delta}\gamma\epsilon_R^{\pi^{k+1}}}{(1-\gamma)^2}.$$

Again, we have the following constraint violation bound for each policy update:

$$J^C(\pi^k) + \frac{1}{1-\gamma}\mathbb{E}_{\substack{s \sim d^{\pi^k} \\ a \sim \pi^{k+1}}}\left[A_R^{\pi^k}(s,a)\right] \leq h, \tag{9}$$

and

$$J^C(\pi^{k+1}) - J^C(\pi^k) \leq \frac{1}{1-\gamma} \mathbb{E}_{\substack{s \sim d^{\pi^k} \\ a \sim \pi^{k+1}}} \left[ A_C^{\pi^k}(s,a) + \frac{2\gamma \epsilon_C^{\pi^{k+1}}}{1-\gamma} \sqrt{\frac{1}{2} D_{\mathrm{KL}}(\pi^{k+1}||\pi^k)[s]} \right]. \quad (10)$$

Combining Eq. (9) and Eq. (10), we have

$$\begin{aligned}
J^C(\pi^{k+1}) &\leq h + \frac{1}{1-\gamma} \mathbb{E}_{\substack{s \sim d^{\pi^k} \\ a \sim \pi^{k+1}}} \left[ \frac{2\gamma \epsilon_C^{\pi^{k+1}}}{1-\gamma} \sqrt{\frac{1}{2} D_{\mathrm{KL}}(\pi^{k+1}||\pi^k)[s]} \right] \\
&\leq h + \frac{\sqrt{2\delta}\gamma \epsilon_C^{\pi^{k+1}}}{(1-\gamma)^2}.
\end{aligned}$$

$\square$

## S.2 Proof of Theorem 3.2: Performance Bound on Updating the Constraint-violating Policy

To prove the policy performance bound when the current policy is infeasible (*i.e.,* constraint-violating), we prove the KL divergence between $\pi^k$ and $\pi^{k+1}$ for the KL divergence projection. We then prove our main theorem for the worst-case performance degradation.

**Lemma S.3.** *If the current policy $\pi^k$ violates the constraint, the constraint set is closed and convex, the KL divergence constraint for the first step is $\mathbb{E}_{s \sim d^{\pi^k}}\left[ D_{\mathrm{KL}}(\pi^{k+\frac{1}{2}}||\pi^k)[s] \right] \leq \delta$, where $\delta$ is the step size in the reward improvement step, then under the KL divergence projection, we have*

$$\mathbb{E}_{s \sim d^{\pi^k}}\left[ D_{\mathrm{KL}}(\pi^{k+1}||\pi^k)[s] \right] \leq \delta + {b^+}^2 \alpha_{\mathrm{KL}},$$

*where $\alpha_{\mathrm{KL}} \doteq \frac{1}{2a^T H^{-1} a}$, $a$ is the gradient of the cost advantage function, $H$ is the Hessian of the KL divergence constraint, and $b^+ \doteq \max(0, J^C(\pi^k) - h)$.*

*Proof.* We define the sublevel set of cost constraint function for the current infeasible policy $\pi^k$:

$$L^{\pi^k} = \{ \pi \mid J^C(\pi^k) + \mathbb{E}_{\substack{s \sim d^{\pi^k} \\ a \sim \pi}} [A_C^{\pi^k}(s,a)] \leq J^C(\pi^k) \}.$$

This implies that the current policy $\pi^k$ lies in $L^{\pi^k}$, and $\pi^{k+\frac{1}{2}}$ is projected onto the constraint set: $\{ \pi \mid J^C(\pi^k) + \mathbb{E}_{\substack{s \sim d^{\pi^k} \\ a \sim \pi}} [A_C^{\pi^k}(s,a)] \leq h \}$. Next, we define the policy $\pi_l^{k+1}$ as the projection of $\pi^{k+\frac{1}{2}}$ onto $L^{\pi^k}$.

By the Three-point Lemma, for these three polices $\pi^k, \pi^{k+1}$, and $\pi_l^{k+1}$, with $\varphi(\boldsymbol{x}) \doteq \sum_i x_i \log x_i$ (this is illustrated in Fig. 6), we have

$$\begin{aligned}
\delta \geq \mathbb{E}_{s \sim d^{\pi^k}}\left[ D_{\mathrm{KL}}(\pi_l^{k+1}||\pi^k)[s] \right] = {} & \mathbb{E}_{s \sim d^{\pi^k}}\left[ D_{\mathrm{KL}}(\pi^{k+1}||\pi^k)[s] \right] \\
& - \mathbb{E}_{s \sim d^{\pi^k}}\left[ D_{\mathrm{KL}}(\pi^{k+1}||\pi_l^{k+1})[s] \right] \\
& + \mathbb{E}_{s \sim d^{\pi^k}}\left[ (\nabla \varphi(\pi^k) - \nabla \varphi(\pi_l^{k+1}))^T (\pi^{k+1} - \pi_l^{k+1})[s] \right] \\
\Rightarrow \mathbb{E}_{s \sim d^{\pi^k}}\left[ D_{\mathrm{KL}}(\pi^{k+1}||\pi^k)[s] \right] \leq {} & \delta + \mathbb{E}_{s \sim d^{\pi^k}}\left[ D_{\mathrm{KL}}(\pi^{k+1}||\pi_l^{k+1})[s] \right] \\
& - \mathbb{E}_{s \sim d^{\pi^k}}\left[ (\nabla \varphi(\pi^k) - \nabla \varphi(\pi_l^{k+1}))^T (\pi^{k+1} - \pi_l^{k+1})[s] \right]. \quad (11)
\end{aligned}$$

The inequality $\mathbb{E}_{s \sim d^{\pi^k}}\left[ D_{\mathrm{KL}}(\pi_l^{k+1}||\pi^k)[s] \right] \leq \delta$ comes from that $\pi^k$ and $\pi_l^{k+1}$ are in $L^{\pi^k}$, and Lemma S.1.

If the constraint violation of the current policy $\pi^k$ is small, *i.e.,* $b^+$ is small, $\mathbb{E}_{s \sim d^{\pi^k}}\left[ D_{\mathrm{KL}}(\pi^{k+1}||\pi_l^{k+1})[s] \right]$ can be approximated by the second order expansion. By the

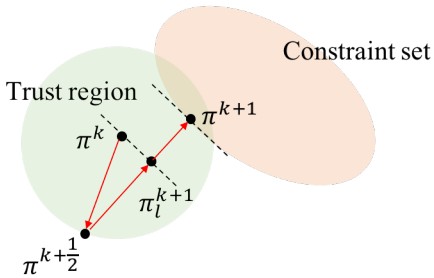

Figure 6: Update procedures for PCPO when the current policy $\pi^k$ is infeasible. $\pi_l^{k+1}$ is the projection of $\pi^{k+\frac{1}{2}}$ onto the sublevel set of the constraint set. We find the KL divergence between $\pi^k$ and $\pi^{k+1}$.

update rule in Eq. (6), we have

$$
\begin{aligned}
\mathbb{E}_{s\sim d^{\pi^k}}\left[D_{\mathrm{KL}}(\pi^{k+1}||\pi_l^{k+1})[s]\right] &\approx \frac{1}{2}(\boldsymbol{\theta}^{k+1}-\boldsymbol{\theta}_l^{k+1})^T\boldsymbol{H}(\boldsymbol{\theta}^{k+1}-\boldsymbol{\theta}_l^{k+1}) \\
&= \frac{1}{2}\left(\frac{b^+}{\boldsymbol{a}^T\boldsymbol{H}^{-1}\boldsymbol{a}}\boldsymbol{H}^{-1}\boldsymbol{a}\right)^T\boldsymbol{H}\left(\frac{b^+}{\boldsymbol{a}^T\boldsymbol{H}^{-1}\boldsymbol{a}}\boldsymbol{H}^{-1}\boldsymbol{a}\right) \\
&= \frac{b^{+2}}{2\boldsymbol{a}^T\boldsymbol{H}^{-1}\boldsymbol{a}} \\
&= b^{+2}\alpha_{\mathrm{KL}},
\end{aligned} \tag{12}
$$

where $\alpha_{\mathrm{KL}} \doteq \frac{1}{2\boldsymbol{a}^T\boldsymbol{H}^{-1}\boldsymbol{a}}$.

And since $\delta$ is small, we have $\nabla\varphi(\pi^k) - \nabla\varphi(\pi_l^{k+1}) \approx \boldsymbol{0}$ given $s$. Thus, the third term in Eq. (11) can be eliminated.

Combining Eq. (11) and Eq. (12), we have

$$
\mathbb{E}_{s\sim d^{\pi^k}}\left[D_{\mathrm{KL}}(\pi^{k+1}||\pi^k)[s]\right] \le \delta + b^{+2}\alpha_{\mathrm{KL}}.
$$

$\square$

Now we use Lemma S.3 to prove our main theorem.

**Theorem S.4.** *Define* $\epsilon_R^{\pi^{k+1}} \doteq \max_s\left|\mathbb{E}_{a\sim\pi^{k+1}}[A_R^{\pi^k}(s,a)]\right|$, $\epsilon_C^{\pi^{k+1}} \doteq \max_s\left|\mathbb{E}_{a\sim\pi^{k+1}}[A_C^{\pi^k}(s,a)]\right|$, $b^+ \doteq \max(0, J^C(\pi^k)-h)$, *and* $\alpha_{\mathrm{KL}} \doteq \frac{1}{2\boldsymbol{a}^T\boldsymbol{H}^{-1}\boldsymbol{a}}$, *where* $\boldsymbol{a}$ *is the gradient of the cost advantage function and* $\boldsymbol{H}$ *is the Hessian of the KL divergence constraint. If the current policy* $\pi^k$ *violates the constraint, then under the KL divergence projection, the lower bound on reward improvement and the upper bound on constraint violation for each policy update are*

$$
J^R(\pi^{k+1}) - J^R(\pi^k) \ge -\frac{\sqrt{2(\delta + b^{+2}\alpha_{\mathrm{KL}})}\gamma\epsilon_R^{\pi^{k+1}}}{(1-\gamma)^2},
$$

$$
\text{and } J^C(\pi^{k+1}) \le h + \frac{\sqrt{2(\delta + b^{+2}\alpha_{\mathrm{KL}})}\gamma\epsilon_C^{\pi^{k+1}}}{(1-\gamma)^2},
$$

*where* $\delta$ *is the step size in the reward improvement step.*

*Proof.* Following the same proof in Theorem S.2, we complete the proof. $\square$

Note that the bounds we obtain for the infeasibe case; to the best of our knowledge, are new results.

## S.3 PROOF OF ANALYTICAL SOLUTION TO PCPO

**Theorem S.5.** *Consider the PCPO problem. In the first step, we optimize the reward:*

$$\boldsymbol{\theta}^{k+\frac{1}{2}} = \arg\max_{\boldsymbol{\theta}} \quad \boldsymbol{g}^T(\boldsymbol{\theta} - \boldsymbol{\theta}^k)$$

$$s.t. \quad \frac{1}{2}(\boldsymbol{\theta} - \boldsymbol{\theta}^k)^T \boldsymbol{H}(\boldsymbol{\theta} - \boldsymbol{\theta}^k) \leq \delta,$$

*and in the second step, we project the policy onto the constraint set:*

$$\boldsymbol{\theta}^{k+1} = \arg\min_{\boldsymbol{\theta}} \quad \frac{1}{2}(\boldsymbol{\theta} - \boldsymbol{\theta}^{k+\frac{1}{2}})^T \boldsymbol{L}(\boldsymbol{\theta} - \boldsymbol{\theta}^{k+\frac{1}{2}})$$

$$s.t. \quad \boldsymbol{a}^T(\boldsymbol{\theta} - \boldsymbol{\theta}^k) + b \leq 0,$$

*where $\boldsymbol{g}, \boldsymbol{a}, \boldsymbol{\theta} \in \mathbb{R}^n, b, \delta \in \mathbb{R}, \delta > 0$, and $\boldsymbol{H}, \boldsymbol{L} \in \mathbb{R}^{n \times n}, \boldsymbol{L} = \boldsymbol{H}$ if using the KL divergence projection, and $\boldsymbol{L} = \boldsymbol{I}$ if using the $L^2$ norm projection. When there is at least one strictly feasible point, the optimal solution satisfies*

$$\boldsymbol{\theta}^{k+1} = \boldsymbol{\theta}^k + \sqrt{\frac{2\delta}{\boldsymbol{g}^T \boldsymbol{H}^{-1} \boldsymbol{g}}} \boldsymbol{H}^{-1}\boldsymbol{g} - \max(0, \frac{\sqrt{\frac{2\delta}{\boldsymbol{g}^T \boldsymbol{H}^{-1} \boldsymbol{g}}} \boldsymbol{a}^T \boldsymbol{H}^{-1}\boldsymbol{g} + b}{\boldsymbol{a}^T \boldsymbol{L}^{-1}\boldsymbol{a}}) \boldsymbol{L}^{-1}\boldsymbol{a},$$

*assuming that $\boldsymbol{H}$ is invertible to get a unique solution.*

*Proof.* For the first problem, since $\boldsymbol{H}$ is the Fisher Information matrix, which automatically guarantees it is positive semi-definite. Hence it is a convex program with quadratic inequality constraints. Hence if the primal problem has a feasible point, then Slaters condition is satisfied and strong duality holds. Let $\boldsymbol{\theta}^*$ and $\lambda^*$ denote the solutions to the primal and dual problems, respectively. In addition, the primal objective function is continuously differentiable. Hence the Karush-Kuhn-Tucker (KKT) conditions are necessary and sufficient for the optimality of $\boldsymbol{\theta}^*$ and $\lambda^*$. We now form the Lagrangian:

$$\mathcal{L}(\boldsymbol{\theta}, \lambda) = -\boldsymbol{g}^T(\boldsymbol{\theta} - \boldsymbol{\theta}^k) + \lambda\Big(\frac{1}{2}(\boldsymbol{\theta} - \boldsymbol{\theta}^k)^T \boldsymbol{H}(\boldsymbol{\theta} - \boldsymbol{\theta}^k) - \delta\Big).$$

And we have the following KKT conditions:

$$-\boldsymbol{g} + \lambda^* \boldsymbol{H}\boldsymbol{\theta}^* - \lambda^* \boldsymbol{H}\boldsymbol{\theta}^k = 0 \qquad \nabla_{\boldsymbol{\theta}}\mathcal{L}(\boldsymbol{\theta}^*, \lambda^*) = 0 \tag{13}$$

$$\frac{1}{2}(\boldsymbol{\theta}^* - \boldsymbol{\theta}^k)^T \boldsymbol{H}(\boldsymbol{\theta}^* - \boldsymbol{\theta}^k) - \delta = 0 \qquad \nabla_{\lambda}\mathcal{L}(\boldsymbol{\theta}^*, \lambda^*) = 0 \tag{14}$$

$$\frac{1}{2}(\boldsymbol{\theta}^* - \boldsymbol{\theta}^k)^T \boldsymbol{H}(\boldsymbol{\theta}^* - \boldsymbol{\theta}^k) - \delta \leq 0 \qquad \text{primal constraints} \tag{15}$$

$$\lambda^* \geq 0 \qquad \text{dual constraints} \tag{16}$$

$$\lambda^*\Big(\frac{1}{2}(\boldsymbol{\theta}^* - \boldsymbol{\theta}^k)^T \boldsymbol{H}(\boldsymbol{\theta}^* - \boldsymbol{\theta}^k) - \delta\Big) = 0 \qquad \text{complementary slackness} \tag{17}$$

By Eq. (13), we have $\boldsymbol{\theta}^* = \boldsymbol{\theta}^k + \frac{1}{\lambda^*}\boldsymbol{H}^{-1}\boldsymbol{g}$. And by plugging Eq. (13) into Eq. (14), we have $\lambda^* = \sqrt{\frac{\boldsymbol{g}^T \boldsymbol{H}^{-1}\boldsymbol{g}}{2\delta}}$. Hence we have our optimal solution:

$$\boldsymbol{\theta}^{k+\frac{1}{2}} = \boldsymbol{\theta}^* = \boldsymbol{\theta}^k + \sqrt{\frac{2\delta}{\boldsymbol{g}^T \boldsymbol{H}^{-1}\boldsymbol{g}}} \boldsymbol{H}^{-1}\boldsymbol{g}, \tag{18}$$

which also satisfies Eq. (15), Eq. (16), and Eq. (17).

Following the same reasoning, we now form the Lagrangian of the second problem:

$$\mathcal{L}(\boldsymbol{\theta}, \lambda) = \frac{1}{2}(\boldsymbol{\theta} - \boldsymbol{\theta}^{k+\frac{1}{2}})^T \boldsymbol{L}(\boldsymbol{\theta} - \boldsymbol{\theta}^{k+\frac{1}{2}}) + \lambda(\boldsymbol{a}^T(\boldsymbol{\theta} - \boldsymbol{\theta}^k) + b).$$

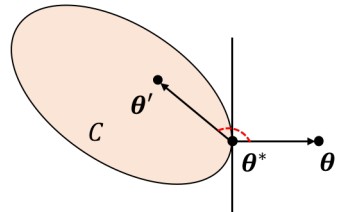

Figure 7: The projection onto the convex set with $\boldsymbol{\theta}' \in \mathcal{C}$ and $\boldsymbol{\theta}^* = \mathrm{Proj}_{\mathcal{C}}^{L}(\boldsymbol{\theta})$.

And we have the following KKT conditions:

$$\boldsymbol{L}\boldsymbol{\theta}^* - \boldsymbol{L}\boldsymbol{\theta}^{k+\frac{1}{2}} + \lambda^* \boldsymbol{a} = 0 \quad \nabla_{\boldsymbol{\theta}} \mathcal{L}(\boldsymbol{\theta}^*, \lambda^*) = 0 \tag{19}$$

$$\boldsymbol{a}^T(\boldsymbol{\theta}^* - \boldsymbol{\theta}^k) + b = 0 \quad \nabla_{\lambda} \mathcal{L}(\boldsymbol{\theta}^*, \lambda^*) = 0 \tag{20}$$

$$\boldsymbol{a}^T(\boldsymbol{\theta}^* - \boldsymbol{\theta}^k) + b \le 0 \quad \text{primal constraints} \tag{21}$$

$$\lambda^* \ge 0 \quad \text{dual constraints} \tag{22}$$

$$\lambda^*(\boldsymbol{a}^T(\boldsymbol{\theta}^* - \boldsymbol{\theta}^k) + b) = 0 \quad \text{complementary slackness} \tag{23}$$

By Eq. (19), we have $\boldsymbol{\theta}^* = \boldsymbol{\theta}^{k+1} + \lambda^* \boldsymbol{L}^{-1} \boldsymbol{a}$. And by plugging Eq. (19) into Eq. (20) and Eq. (22), we have $\lambda^* = \max(0, \frac{\boldsymbol{a}^T(\boldsymbol{\theta}^{k+\frac{1}{2}} - \boldsymbol{\theta}^k) + b}{\boldsymbol{a} \boldsymbol{L}^{-1} \boldsymbol{a}})$. Hence we have our optimal solution:

$$\boldsymbol{\theta}^{k+1} = \boldsymbol{\theta}^* = \boldsymbol{\theta}^{k+\frac{1}{2}} - \max(0, \frac{\boldsymbol{a}^T(\boldsymbol{\theta}^{k+\frac{1}{2}} - \boldsymbol{\theta}^k) + b}{\boldsymbol{a}^T \boldsymbol{L}^{-1} \boldsymbol{a}^T}) \boldsymbol{L}^{-1} \boldsymbol{a}, \tag{24}$$

which also satisfies Eq. (21) and Eq. (23). Hence by Eq. (18) and Eq. (24), we have

$$\boldsymbol{\theta}^{k+1} = \boldsymbol{\theta}^k + \sqrt{\frac{2\delta}{\boldsymbol{g}^T \boldsymbol{H}^{-1} \boldsymbol{g}}} \boldsymbol{H}^{-1} \boldsymbol{g} - \max(0, \frac{\sqrt{\frac{2\delta}{\boldsymbol{g}^T \boldsymbol{H}^{-1} \boldsymbol{g}}} \boldsymbol{a}^T \boldsymbol{H}^{-1} \boldsymbol{g} + b}{\boldsymbol{a}^T \boldsymbol{L}^{-1} \boldsymbol{a}}) \boldsymbol{L}^{-1} \boldsymbol{a}.$$

$\square$

## S.4 PROOF OF THEOREM 4.1: STATIONARY POINTS OF PCPO WITH THE KL DIVERGENCE AND $L^2$ NORM PROJECTIONS

For our analysis, we make the following assumptions: we *minimize* the negative reward objective function $f : \mathbb{R}^n \to \mathbb{R}$ (We follow the convention of the literature that authors typically minimize the objective function). The function $f$ is $L$-smooth and twice continuously differentiable over the closed and convex constraint set $\mathcal{C}$. We have the following lemma to characterize the projection and for the proof of Theorem S.7. (See Fig. 7 for semantic illustration.)

**Lemma S.6.** *For any* $\boldsymbol{\theta}$, $\boldsymbol{\theta}^* = \mathrm{Proj}_{\mathcal{C}}^{L}(\boldsymbol{\theta})$ *if and only if* $(\boldsymbol{\theta} - \boldsymbol{\theta}^*)^T \boldsymbol{L}(\boldsymbol{\theta}' - \boldsymbol{\theta}^*) \le 0, \forall \boldsymbol{\theta}' \in \mathcal{C}$, *where* $\mathrm{Proj}_{\mathcal{C}}^{L}(\boldsymbol{\theta}) \doteq \arg\min_{\boldsymbol{\theta}' \in \mathcal{C}} ||\boldsymbol{\theta} - \boldsymbol{\theta}'||_L^2$, *and* $\boldsymbol{L} = \boldsymbol{H}$ *if using the KL divergence projection, and* $\boldsymbol{L} = \boldsymbol{I}$ *if using the* $L^2$ *norm projection.*

*Proof.* ($\Rightarrow$) Let $\boldsymbol{\theta}^* = \mathrm{Proj}_{\mathcal{C}}^{L}(\boldsymbol{\theta})$ for a given $\boldsymbol{\theta} \notin \mathcal{C}$, $\boldsymbol{\theta}' \in \mathcal{C}$ be such that $\boldsymbol{\theta}' \ne \boldsymbol{\theta}^*$, and $\alpha \in (0, 1)$. Then we have

$$||\boldsymbol{\theta} - \boldsymbol{\theta}^*||_L^2 \le ||\boldsymbol{\theta} - (\boldsymbol{\theta}^* + \alpha(\boldsymbol{\theta}' - \boldsymbol{\theta}^*))||_L^2$$

$$= ||\boldsymbol{\theta} - \boldsymbol{\theta}^*||_L^2 + \alpha^2 ||\boldsymbol{\theta}' - \boldsymbol{\theta}^*||_L^2 - 2\alpha(\boldsymbol{\theta} - \boldsymbol{\theta}^*)^T \boldsymbol{L}(\boldsymbol{\theta}' - \boldsymbol{\theta}^*)$$

$$\Rightarrow (\boldsymbol{\theta} - \boldsymbol{\theta}^*)^T \boldsymbol{L}(\boldsymbol{\theta}' - \boldsymbol{\theta}^*) \le \frac{\alpha}{2} ||\boldsymbol{\theta}' - \boldsymbol{\theta}^*||_L^2. \tag{25}$$

Since the right hand side of Eq. (25) can be made arbitrarily small for a given $\alpha$, and hence we have:

$$(\boldsymbol{\theta} - \boldsymbol{\theta}^*)^T \boldsymbol{L}(\boldsymbol{\theta}' - \boldsymbol{\theta}^*) \le 0, \forall \boldsymbol{\theta}' \in \mathcal{C}.$$

($\Leftarrow$) Let $\boldsymbol{\theta}^* \in \mathcal{C}$ be such that $(\boldsymbol{\theta} - \boldsymbol{\theta}^*)^T \boldsymbol{L}(\boldsymbol{\theta}' - \boldsymbol{\theta}^*) \leq 0, \forall \theta' \in \mathcal{C}$. We show that $\boldsymbol{\theta}^*$ must be the optimal solution. Let $\boldsymbol{\theta}' \in \mathcal{C}$ and $\boldsymbol{\theta}' \neq \boldsymbol{\theta}^*$. Then we have

$$
\begin{aligned}
||\boldsymbol{\theta} - \boldsymbol{\theta}'||_{\boldsymbol{L}}^2 - ||\boldsymbol{\theta} - \boldsymbol{\theta}^*||_{\boldsymbol{L}}^2 &= ||\boldsymbol{\theta} - \boldsymbol{\theta}^* + \boldsymbol{\theta}^* - \boldsymbol{\theta}'||_{\boldsymbol{L}}^2 - ||\boldsymbol{\theta} - \boldsymbol{\theta}^*||_{\boldsymbol{L}}^2 \\
&= ||\boldsymbol{\theta} - \boldsymbol{\theta}^*||_{\boldsymbol{L}}^2 + ||\boldsymbol{\theta}' - \boldsymbol{\theta}^*||_{\boldsymbol{L}}^2 - 2(\boldsymbol{\theta} - \boldsymbol{\theta}^*)^T \boldsymbol{L}(\boldsymbol{\theta}' - \boldsymbol{\theta}^*) - ||\boldsymbol{\theta} - \boldsymbol{\theta}^*||_{\boldsymbol{L}}^2 \\
&> 0 \\
\Rightarrow ||\boldsymbol{\theta} - \boldsymbol{\theta}'||_{\boldsymbol{L}}^2 &> ||\boldsymbol{\theta} - \boldsymbol{\theta}^*||_{\boldsymbol{L}}^2.
\end{aligned}
$$

Hence, $\boldsymbol{\theta}^*$ is the optimal solution to the optimization problem, and $\boldsymbol{\theta}^* = \mathrm{Proj}_{\mathcal{C}}^{\boldsymbol{L}}(\boldsymbol{\theta})$. $\square$

Based on Lemma S.6, we have the following theorem.

**Theorem S.7.** *Let* $\eta \doteq \sqrt{\frac{2\delta}{\boldsymbol{g}^T \boldsymbol{H}^{-1} \boldsymbol{g}}}$ *in Eq. (6), where $\delta$ is the step size for reward improvement, $\boldsymbol{g}$ is the gradient of $f$, $\boldsymbol{H}$ is the Fisher information matrix. Let $\sigma_{\max}(\boldsymbol{H})$ be the largest singular value of $\boldsymbol{H}$, and $\boldsymbol{a}$ be the gradient of cost advantage function in Eq. (6). Then PCPO with the KL divergence projection converges to stationary points with $\boldsymbol{g} \in -\boldsymbol{a}$ (i.e., the gradient of $f$ belongs to the negative gradient of the cost advantage function). The objective value changes by*

$$ f(\boldsymbol{\theta}^{k+1}) \leq f(\boldsymbol{\theta}^k) + ||\boldsymbol{\theta}^{k+1} - \boldsymbol{\theta}^k||^2_{-\frac{1}{\eta}\boldsymbol{H} + \frac{L}{2}\boldsymbol{I}}. \tag{26} $$

*PCPO with the $L^2$ norm projection converges to stationary points with $\boldsymbol{H}^{-1}\boldsymbol{g} \in -\boldsymbol{a}$ (i.e., the product of the inverse of $\boldsymbol{H}$ and gradient of $f$ belongs to the negative gradient of the cost advantage function). If $\sigma_{\max}(\boldsymbol{H}) \leq 1$, then the objective value changes by*

$$ f(\boldsymbol{\theta}^{k+1}) \leq f(\boldsymbol{\theta}^k) + (\frac{L}{2} - \frac{1}{\eta})||\boldsymbol{\theta}^{k+1} - \boldsymbol{\theta}^k||_2^2. \tag{27} $$

*Proof.* The proof of the theorem is based on working in a Hilbert space and the non-expansive property of the projection. We first prove stationary points for PCPO with the KL divergence and $L^2$ norm projections, and then prove the change of the objective value.

When in stationary points $\boldsymbol{\theta}^*$, we have

$$ \boldsymbol{\theta}^* = \boldsymbol{\theta}^* - \sqrt{\frac{2\delta}{\boldsymbol{g}^T \boldsymbol{H}^{-1}\boldsymbol{g}}}\boldsymbol{H}^{-1}\boldsymbol{g} - \max(0, \frac{\sqrt{\frac{2\delta}{\boldsymbol{g}^T\boldsymbol{H}^{-1}\boldsymbol{g}}}\boldsymbol{a}^T\boldsymbol{H}^{-1}\boldsymbol{g} + b}{\boldsymbol{a}^T\boldsymbol{L}^{-1}\boldsymbol{a}})\boldsymbol{L}^{-1}\boldsymbol{a}. $$

$$ \Leftrightarrow \sqrt{\frac{2\delta}{\boldsymbol{g}^T\boldsymbol{H}^{-1}\boldsymbol{g}}}\boldsymbol{H}^{-1}\boldsymbol{g} = -\max(0, \frac{\sqrt{\frac{2\delta}{\boldsymbol{g}^T\boldsymbol{H}^{-1}\boldsymbol{g}}}\boldsymbol{a}^T\boldsymbol{H}^{-1}\boldsymbol{g} + b}{\boldsymbol{a}^T\boldsymbol{L}^{-1}\boldsymbol{a}})\boldsymbol{L}^{-1}\boldsymbol{a} $$

$$ \Leftrightarrow \boldsymbol{H}^{-1}\boldsymbol{g} \in -\boldsymbol{L}^{-1}\boldsymbol{a}. \tag{28} $$

For the KL divergence projection ($\boldsymbol{L} = \boldsymbol{H}$), Eq. (28) boils down to $\boldsymbol{g} \in -\boldsymbol{a}$, and for the $L^2$ norm projection ($\boldsymbol{L} = \boldsymbol{I}$), Eq. (28) is equivalent to $\boldsymbol{H}^{-1}\boldsymbol{g} \in -\boldsymbol{a}$.

Now we prove the second part of the theorem. Based on Lemma S.6, for the KL divergence projection, we have

$$ (\boldsymbol{\theta}^k - \boldsymbol{\theta}^{k+1})^T \boldsymbol{H}(\boldsymbol{\theta}^k - \eta\boldsymbol{H}^{-1}\boldsymbol{g} - \boldsymbol{\theta}^{k+1}) \leq 0 $$

$$ \Rightarrow \boldsymbol{g}^T(\boldsymbol{\theta}^{k+1} - \boldsymbol{\theta}^k) \leq -\frac{1}{\eta}||\boldsymbol{\theta}^{k+1} - \boldsymbol{\theta}^k||_{\boldsymbol{H}}^2. \tag{29} $$

By Eq. (29), and $L$-smooth continuous function $f$, we have

$$
\begin{aligned}
f(\boldsymbol{\theta}^{k+1}) &\leq f(\boldsymbol{\theta}^k) + \boldsymbol{g}^T(\boldsymbol{\theta}^{k+1} - \boldsymbol{\theta}^k) + \frac{L}{2}||\boldsymbol{\theta}^{k+1} - \boldsymbol{\theta}^k||_2^2 \\
&\leq f(\boldsymbol{\theta}^k) - \frac{1}{\eta}||\boldsymbol{\theta}^{k+1} - \boldsymbol{\theta}^k||_{\boldsymbol{H}}^2 + \frac{L}{2}||\boldsymbol{\theta}^{k+1} - \boldsymbol{\theta}^k||_2^2 \\
&= f(\boldsymbol{\theta}^k) + (\boldsymbol{\theta}^{k+1} - \boldsymbol{\theta}^k)^T(-\frac{1}{\eta}\boldsymbol{H} + \frac{L}{2}\boldsymbol{I})(\boldsymbol{\theta}^{k+1} - \boldsymbol{\theta}^k) \\
&= f(\boldsymbol{\theta}^k) + ||\boldsymbol{\theta}^{k+1} - \boldsymbol{\theta}^k||_{-\frac{1}{\eta}\boldsymbol{H} + \frac{L}{2}\boldsymbol{I}}^2.
\end{aligned}
$$

For the $L^2$ norm projection, we have

$$(\boldsymbol{\theta}^k - \boldsymbol{\theta}^{k+1})^T(\boldsymbol{\theta}^k - \eta\boldsymbol{H}^{-1}\boldsymbol{g} - \boldsymbol{\theta}^{k+1}) \leq 0$$

$$\Rightarrow \boldsymbol{g}^T\boldsymbol{H}^{-1}(\boldsymbol{\theta}^{k+1} - \boldsymbol{\theta}^k) \leq -\frac{1}{\eta}||\boldsymbol{\theta}^{k+1} - \boldsymbol{\theta}^k||_2^2. \tag{30}$$

By Eq. (30), $L$-smooth continuous function $f$, and if $\sigma_{\max}(\boldsymbol{H}) \leq 1$, we have

$$f(\boldsymbol{\theta}^{k+1}) \leq f(\boldsymbol{\theta}^k) + \boldsymbol{g}^T(\boldsymbol{\theta}^{k+1} - \boldsymbol{\theta}^k) + \frac{L}{2}||\boldsymbol{\theta}^{k+1} - \boldsymbol{\theta}^k||_2^2$$

$$\leq f(\boldsymbol{\theta}^k) + (\frac{L}{2} - \frac{1}{\eta})||\boldsymbol{\theta}^{k+1} - \boldsymbol{\theta}^k||_2^2.$$

To see why we need the assumption of $\sigma_{\max}(\boldsymbol{H}) \leq 1$, we define $\boldsymbol{H} = \boldsymbol{U}\boldsymbol{\Sigma}\boldsymbol{U}^T$ as the singular value decomposition of $\boldsymbol{H}$ with $\boldsymbol{u}_i$ being the column vector of $\boldsymbol{U}$. Then we have

$$\boldsymbol{g}^T\boldsymbol{H}^{-1}(\boldsymbol{\theta}^{k+1} - \boldsymbol{\theta}^k) = \boldsymbol{g}^T\boldsymbol{U}\boldsymbol{\Sigma}^{-1}\boldsymbol{U}^T(\boldsymbol{\theta}^{k+1} - \boldsymbol{\theta}^k)$$

$$= \boldsymbol{g}^T(\sum_i \frac{1}{\sigma_i(\boldsymbol{H})}\boldsymbol{u}_i\boldsymbol{u}_i^T)(\boldsymbol{\theta}^{k+1} - \boldsymbol{\theta}^k)$$

$$= \sum_i \frac{1}{\sigma_i(\boldsymbol{H})}\boldsymbol{g}^T(\boldsymbol{\theta}^{k+1} - \boldsymbol{\theta}^k).$$

If we want to have

$$\boldsymbol{g}^T(\boldsymbol{\theta}^{k+1} - \boldsymbol{\theta}^k) \leq \boldsymbol{g}^T\boldsymbol{H}^{-1}(\boldsymbol{\theta}^{k+1} - \boldsymbol{\theta}^k) \leq -\frac{1}{\eta}||\boldsymbol{\theta}^{k+1} - \boldsymbol{\theta}^k||_2^2,$$

then every singular value $\sigma_i(\boldsymbol{H})$ of $\boldsymbol{H}$ needs to be smaller than 1, and hence $\sigma_{\max}(\boldsymbol{H}) \leq 1$, which justifies the assumption we use to prove the bound. $\qquad\square$

To make the objective value for PCPO with the KL divergence projection improves, the right hand side of Eq. (26) needs to be negative. Hence we have $\frac{L\eta}{2}\boldsymbol{I} \prec \boldsymbol{H}$, implying that $\sigma_{\min}(\boldsymbol{H}) > \frac{L\eta}{2}$. And to make the objective value for PCPO with the $L^2$ norm projection improves, the right hand side of Eq. (27) needs to be negative. Hence we have $\eta < \frac{2}{L}$, implying that

$$\eta = \sqrt{\frac{2\delta}{\boldsymbol{g}^T\boldsymbol{H}^{-1}\boldsymbol{g}}} < \frac{2}{L}$$

$$\Rightarrow \frac{2\delta}{\boldsymbol{g}^T\boldsymbol{H}^{-1}\boldsymbol{g}} < \frac{4}{L^2}$$

$$\Rightarrow \frac{\boldsymbol{g}^T\boldsymbol{H}^{-1}\boldsymbol{g}}{2\delta} > \frac{L^2}{4}$$

$$\Rightarrow \frac{L^2\delta}{2} < \boldsymbol{g}^T\boldsymbol{H}^{-1}\boldsymbol{g}$$

$$\leq ||\boldsymbol{g}||_2||\boldsymbol{H}^{-1}\boldsymbol{g}||_2$$

$$\leq ||\boldsymbol{g}||_2||\boldsymbol{H}^{-1}||_2||\boldsymbol{g}||_2$$

$$= \sigma_{\max}(\boldsymbol{H}^{-1})||\boldsymbol{g}||_2^2$$

$$= \sigma_{\min}(\boldsymbol{H})||\boldsymbol{g}||_2^2$$

$$\Rightarrow \sigma_{\min}(\boldsymbol{H}) > \frac{L^2\delta}{2||\boldsymbol{g}||_2^2}. \tag{31}$$

By the definition of the condition number and Eq. (31), we have

$$\frac{1}{\sigma_{\min}(\boldsymbol{H})} < \frac{2||\boldsymbol{g}||_2^2}{L^2\delta}$$

$$\Rightarrow \frac{\sigma_{\max}(\boldsymbol{H})}{\sigma_{\min}(\boldsymbol{H})} < \frac{2||\boldsymbol{g}||_2^2\sigma_{\max}(\boldsymbol{H})}{L^2\delta}$$

$$\leq \frac{2||\boldsymbol{g}||_2^2}{L^2\delta},$$

which justifies what we discuss.

## S.5 Additional Computational Experiments

### S.5.1 Implementation Details

For detailed explanation of the task in Achiam et al. (2017), please refer to the appendix of Achiam et al. (2017). For detailed explanation of the task in Vinitsky et al. (2018), please refer to Vinitsky et al. (2018).

We use neural networks that take the input of state, and output the mean and variance to be the Gaussian policy in all experiments. For the simulations in the gather and circle tasks, we use a neural network with two hidden layers of size $(64, 32)$. For the simulations in the grid and bottleneck tasks, we use a neural network with two hidden layers of size $(16, 16)$ and $(50, 25)$, respectively. We use $\tanh$ as the activation function of the neural network.

We use GAE-$\lambda$ approach (Schulman et al., 2015b) to estimate $A_R^\pi(s, a)$ and $A_C^\pi(s, a)$. For the simulations in the gather and circle tasks, we use neural network baselines with the same architecture and activation functions as the policy networks. For the simulations in the grid and bottleneck tasks, we use linear baselines.

The hyperparameters of each task for all algorithms are as follows (PC: point circle, PG: point gather, AC: ant circle, AG: ant gather, Gr: grid, and BN: bottleneck tasks):

| Parameter | PC | PG | AC | AG | Gr | BN |
|---|---|---|---|---|---|---|
| discount factor $\gamma$ | 0.995 | 0.995 | 0.995 | 0.995 | 0.999 | 0.999 |
| step size $\delta$ | $10^{-4}$ | $10^{-4}$ | $10^{-4}$ | $10^{-4}$ | $10^{-4}$ | $10^{-4}$ |
| $\lambda_R^{\text{GAE}}$ | 0.95 | 0.95 | 0.95 | 0.95 | 0.97 | 0.97 |
| $\lambda_C^{\text{GAE}}$ | 1.0 | 1.0 | 0.5 | 0.5 | 0.5 | 1.0 |
| Batch size | 50,000 | 50,000 | 100,000 | 100,000 | 10,000 | 25,000 |
| Rollout length | 50 | 15 | 500 | 500 | 400 | 500 |
| Cost constraint threshold $h$ | 5 | 0.1 | 10 | 0.2 | 0 | 0 |

Note that we do not use a learned model to predict the probability of entering an undesirable state within a fixed time horizon as CPO did for cost shaping.

### S.5.2 Experiment Results

To examine the performance of the algorithms with different metrics, we provide the learning curves of the cumulative constraint value over policy update, and the reward versus the cumulative constraint value for the tested algorithms and task pairs in Section 6 shown in Fig. 8. The second metric enables us to compare the reward difference under the same number of cumulative constraint violation.

Overall, we find that,

(a) CPO has more cumulative constraint violation than PCPO.

(b) PCPO with $L^2$ norm projection has less cumulative constraint violation than KL divergence projection except for the point circle and point gather tasks. This observation suggests that the Fisher information matrix is not well-estimated in the high dimensional policy space, leading to have more constraint violation.

(c) PCPO has more reward improvement compared to CPO under the same number of cumulative constraint violation in point circle, point gather, ant circle, ant gather, and bottleneck task.

### S.5.3 CPO without Line Search

Due to approximation errors, CPO performs line search to check whether the updated policy satisfies the trust region and cost constraints. To understand the necessity of line search in CPO, we conducted the experiment with and without line search shown in Fig. 9. The step size $\delta$ is set to 0.01. We find that CPO without line search tends to (1) have large reward variance especially in the point circle task, and (2) learn constraint-satisfying policies slightly faster. These observations

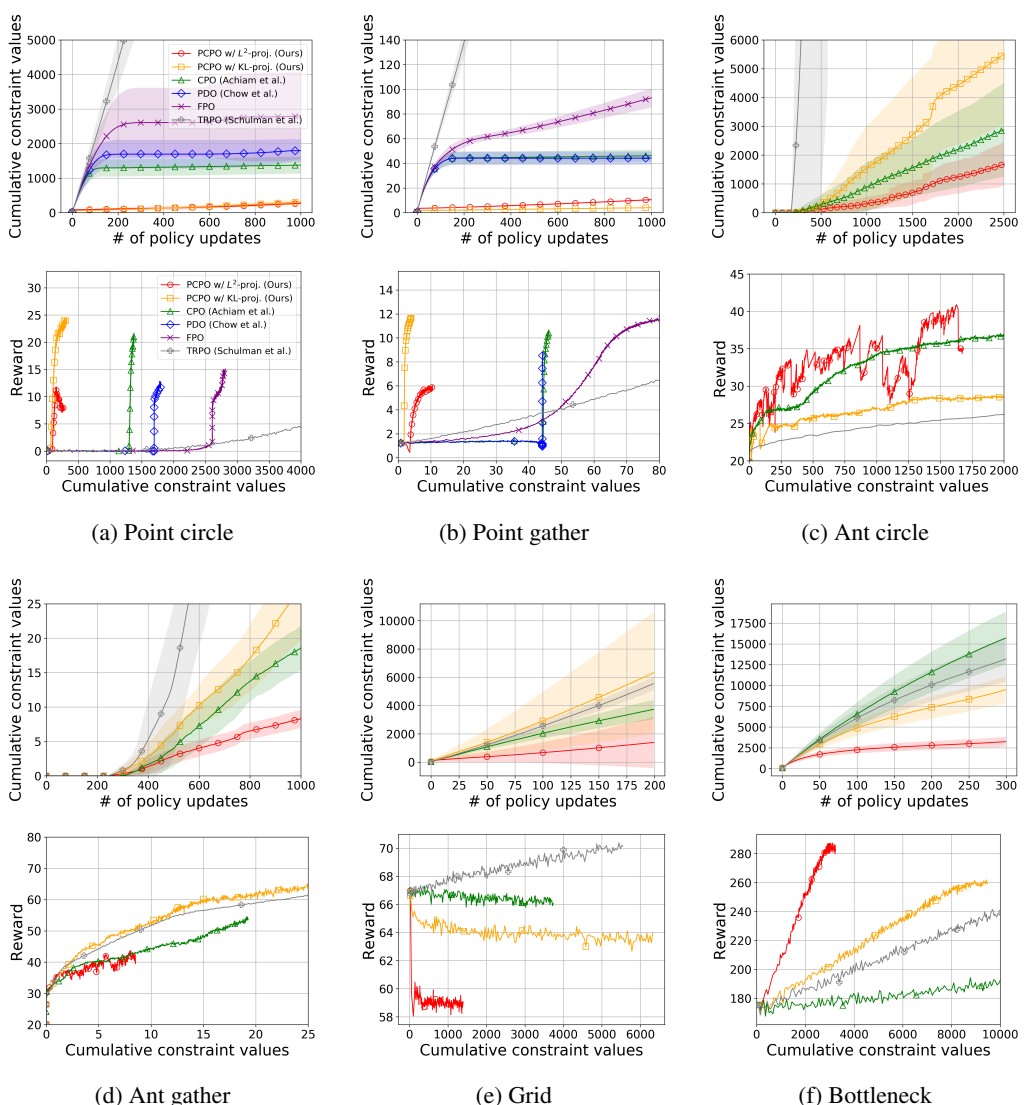

Figure 8: The values of the cumulative constraint value over policy update, and the reward versus the cumulative constraint value for the tested algorithms and task pairs. The solid line is the mean and the shaded area is the standard deviation, over five runs. The curves for baseline oracle, TRPO, indicate the performance when the constraint is ignored. (Best viewed in color, and the legend is shared across all the figures.)

suggest that line search is more conservative in optimizing the policies since it usually take smaller steps. However, we conjecture that if using smaller $\delta$, the effect of line search is not significant.

### S.5.4 THE TASKS WITH HARDER CONSTRAINTS

To understand the stability of PCPO and CPO when deployed in more constraint-critical tasks, we increase the difficulty of the task by setting the constraint threshold to zero and reduce the safe area. The learning curve of discounted reward and constraint value over policy updates are shown in Fig. 10.

We observe that even with more difficult constraint, PCPO still has more reward improvement and constraint satisfaction than CPO, whereas CPO needs more feasible recovery steps to satisfy the constraint. In addition, we observe that PCPO with $L^2$ norm projection has high constraint variance

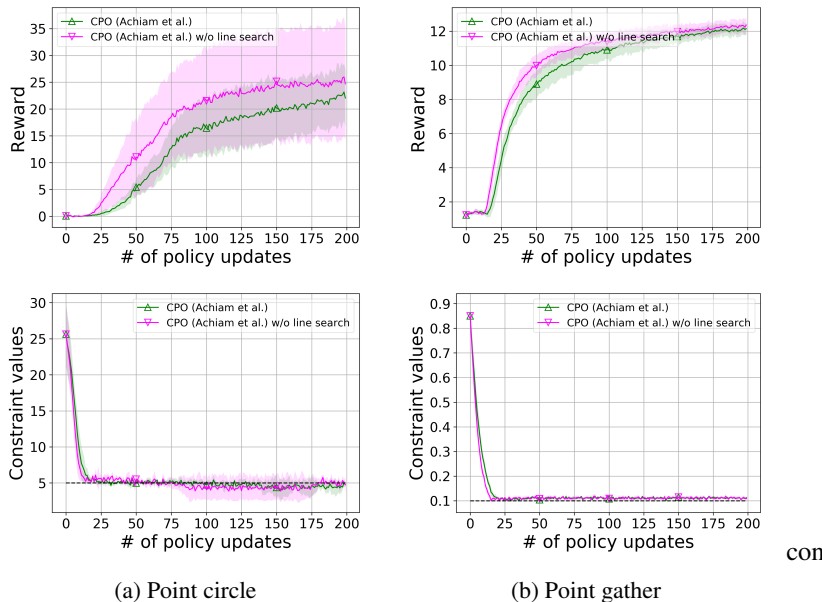

con

(a) Point circle           (b) Point gather

Figure 9: The values of the reward and the constraint value for the tested algorithms and task pairs. The solid line is the mean and the shaded area is the standard deviation, over five runs. The dash line in the cost constraint plot is the cost constraint threshold $h$. Line search helps to stabilize the training. (Best viewed in color)

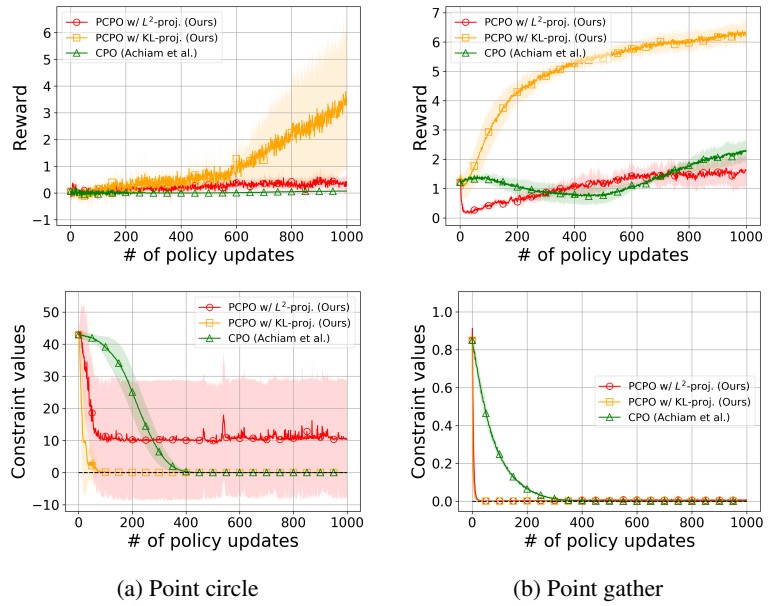

(a) Point circle           (b) Point gather

Figure 10: The values of the reward and the constraint value for the tested algorithms and task pairs. The solid line is the mean and the shaded area is the standard deviation, over five runs. The dash line in the cost constraint plot is the cost constraint threshold $h$. PCPO with KL divergence projection is the only one that can satisfy the constraint with the highest reward. (Best viewed in color)

in point circle task, suggesting that the reward update direction is not well aligned with the cost update direction. We also observe that PCPO with $L^2$ norm projection converges to a bad local

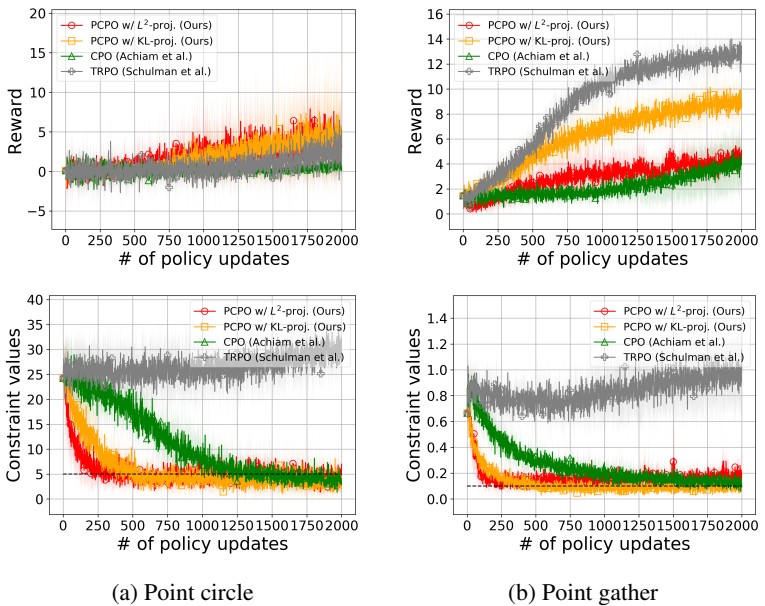

(a) Point circle          (b) Point gather

Figure 11: The values of the reward and the constraint value for the tested algorithms and task pairs. The solid line is the mean and the shaded area is the standard deviation, over five runs. The dash line in the cost constraint plot is the cost constraint threshold $h$. The curves for baseline oracle, TRPO, indicate the reward and constraint violation values when the constraint is ignored. We only use $1\%$ of samples compared to the previous simulations for each policy update. PCPO still satisfies the constraints quickly even when the constraint set is not well-estimated. (Best viewed in color)

optimum in terms of reward in point gather task, suggesting that in order to satisfy the constraint, the cost update direction destroys the reward update direction.

### S.5.5  SMALLER BATCH SAMPLES

To learn policies under constraints, PCPO and CPO require to have a good estimation of the constraint set. However, PCPO may project the policy onto the space that violates the constraint due to the assumption of approximating the constraint set by linear half space constraint. To understand whether the estimation accuracy of the constraint set affects the performance, we conducted the experiments with batch sample size reducing to $1\%$ of the previous experiments (only 500 samples for each policy update) shown in Fig. 11.

We find that smaller training samples affects the performance of the algorithm, creating more reward and cost fluctuation. However, we observe that even with smaller training samples, PCPO still has more reward improvement and constraint satisfaction than CPO.

### S.6  ANALYSIS OF THE APPROXIMATION ERROR AND THE COMPUTATIONAL COST OF THE CONJUGATE GRADIENT METHOD

In the Grid task, we observe that PCPO with KL divergence projection does worse in reward than TRPO, which is expected since TRPO ignores constraints. However, TRPO actually outperforms PCPO with KL divergence projection in terms of constraint, which is unexpected since by trying to consider the constraint, PCPO with KL divergence projection has made constraint satisfaction worse.

The reason for this observation is that the Fisher information matrix is ill-conditioned, *i.e.,* the condition number $\lambda_{\max}(\boldsymbol{H})/\lambda_{\min}(\boldsymbol{H})$ ($\lambda_{\max}$ is the largest eigenvalue of the matrix) of the Fisher information matrix is large, causing conjugate gradient method that computes constraint update direction $\boldsymbol{H}^{-1}\boldsymbol{a}$ with small number of iteration output the inaccurate approximation. Hence the inaccurate

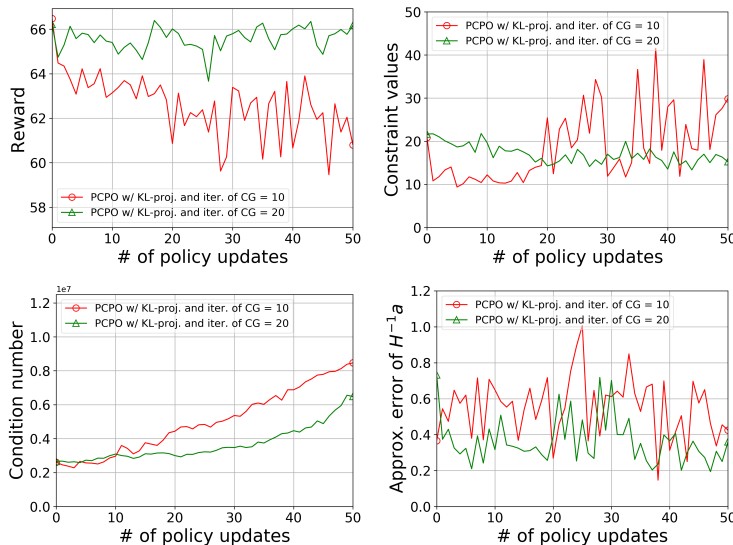

Figure 12: (1) The values of the reward and the constraint, (2) the condition number of the Fisher information matrix, and (3) the approximation error of the constraint update direction over training epochs with the conjugate gradient method's iteration of 10 and 20, respectively. The one with larger number of iteration has more constraint satisfaction since it has more accurate approximation. (Best viewed in color)

approximation of $\boldsymbol{H}^{-1}\boldsymbol{a}$ cause PCPO with KL divergence projection have more constraint violation than TRPO.

To solve this issue, one can have more epochs of conjugate gradient method. This is because that the convergence of conjugate gradient method is controlled by the condition number (Shewchuk, 1994); the larger the condition number is, the more epochs the algorithm needs to get accurate approximation. In our experiments, we set the number of iteration of conjugate gradient method to be 10 to tradeoff between the computational efficiency and the accuracy across all tested algorithms and task pairs.

To verify our observation, we compare the condition number of the Fisher information matrix, and the approximation error of the constraint update direction over training epochs with different number of iteration of the conjugate gradient method shown in Fig. 12.

We observe that the Fisher information matrix is ill-conditioned, and the one with larger number of iteration has less error and more constraint satisfaction. This observation confirms our discussion.

## S.7 COMPARISON OF OPTIMIZATION PATHS OF PCPO WITH KL DIVERGENCE AND $L^2$ NORM PROJECTIONS

Theorem 4.1 states that a stationary point of PCPO with KL divergence projection is different from the one of PCPO with $L^2$ norm projection. See Fig. 13 for illustration. To compare both stationary points, we consider the following example shown in Fig. 14. We *maximize* a non-convex function $f(\boldsymbol{x}) = \boldsymbol{x}^T \text{diag}(\boldsymbol{y})\boldsymbol{x}$ subject to the constraint $\boldsymbol{x}^T \boldsymbol{1} \leq -1$, where $\boldsymbol{y} = [5, -1]^T$, and $\boldsymbol{1}$ is an all-one vector. An optimal solution to this constrained optimization problem is infinity. Fig. 14(a) shows the update direction that combines the objective and the cost constraint update directions for both projections. It shows that PCPO with KL divergence projection has stationary points with $\boldsymbol{g} \in -\boldsymbol{a}$ in the boundary of the constraint set (observe that the update direction is zero for PCPO with KL divergence projection at $\boldsymbol{x} = [0.75, -1.75]^T, [0.25, -1.25]^T$, and $[-0.25, -0.75]^T$), whereas PCPO with $L^2$ norm projection does not have stationary points in the boundary of the constraint set. Furthermore, Fig. 14(b) shows the optimization paths for both projections with one initial starting point. It shows that starting at the initial point $[0.5, -2.0]^T$, PCPO with KL divergence projection with the initial point $[0.5, -2.0]^T$ converges to a local optimum, whereas $L^2$ norm projection converges to

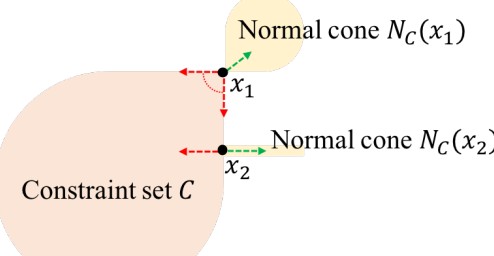

Figure 13: The semantic overview of stationery points of PCPO. The red dashed lines are negative directions of normal cones, and the green dashed lines are objective update directions. The objective update direction in an stationary point is belong to the negative normal cone.

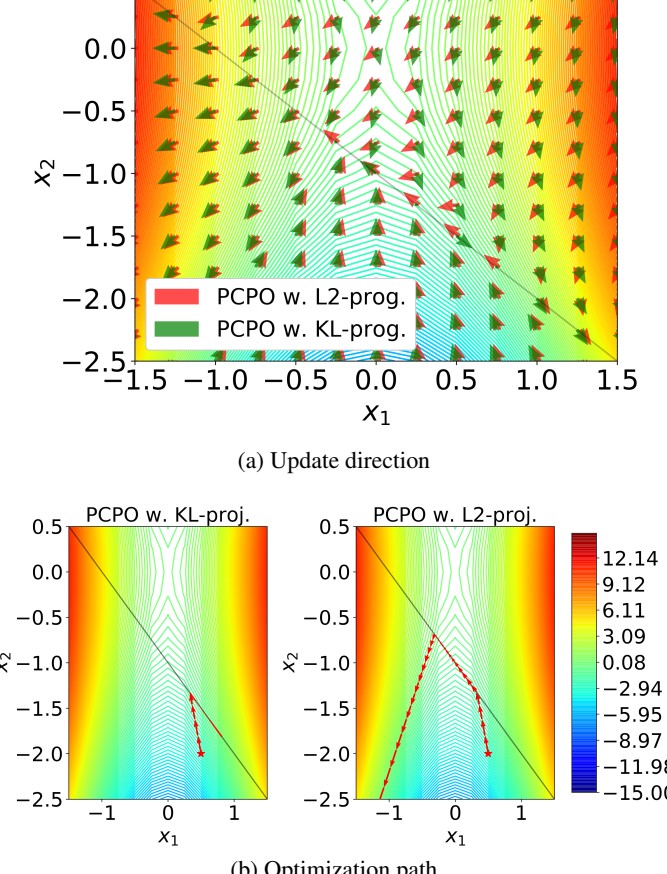

(a) Update direction

(b) Optimization path

Figure 14: The policy update direction that combines the objective and the constraint update directions of each point (top), and the optimization path of PCPO with KL divergence and $L^2$ norm projections with the initial point $[0.5, -2.0]^T$ (below). The red star is the initial point, the red arrows are the optimization paths, and the region that is below to the black line is the constraint set. We see that both projections converge to different solutions.

infinity. However, the above example does not necessary means that PCPO with $L^2$ norm projection always find a better optimum. For example, if the gradient direction of the objective is zero in the constraint set or in the boundary, then both projections may converge to the same stationary point.

