# OpenReview forum: "Projection-Based Constrained Policy Optimization"
_ICLR.cc/2020/Conference — Accept (Poster)_

### Official Review · AnonReviewer1 · 2019-10-16
**Official Blind Review #1**

**Rating:** 6

**Review:**

Summary :

This paper introduces a constrained policy optimization algorithm by introducing a two-step optimization process, where policies that do not satisfy the constraint can be projected back into the constraint set. The proposed PCPO algorithm is theoretically analyzed to provide an upper bound on the constraint violation. The proposed constrained policy optimization algorithm is shown to be useful on a range of control tasks, satisfying constraints or avoiding constraint violation significantly better than the compared baselines.


Comments and Questions :

	- The key idea is to propose an approach to avoid constraint violation in a constrained policy gradient method, where the constraint violation is avoided by first projecting the policy into the constrained set and then choosing a policy from within this set that is guaranteed to satisfy constraints.
	- Existing TRPO method already proposes a constrained optimization method (equation 2 as discussed), where the constraint is within the policy changes. This paper further introduces additional constraints (in the form of expected cost or safety measures) where the intermediary policy from TRPO is further projected into a constraint set and the overall policy improvement is based on the constraint satisfied between the intermediary policy and the improved policy. In other words, there are two levels of constraint satisfaction that is required now for the overall PCPO update.
	- The authors propose two separate distance measures for the secondary constraint update, based on the L2 and KL divergence.
	- I am not sure of the significance of theorem 3.2 in comparison of theorem 3.1? Proof of theorem 3.1 is easy to follow from the appendix, and as noted, follows from Achiam et al., 2017
	- Section 4 discusses similar approximations required as in TRPO for approximating the KL divergence constraint. Similar approximations are requires as in TRPO, with an additional second-order approximation for the KL projection step. The reward improvement step follows similar approximations as in TRPO, and the projection step requires Hessian approximations considering the KL divergence approximation.
	- This seems to be the main bottleneck of the approach? The fact it requires two approximations, mainly for the projection step seems to add further complexity to the propoed approach? The trade-off therefore is to what extent this approximation is required for safe policy improvement in a constrained PO problem versus computational efficiency?
	- Two baselines are mainly used for comparison of results, mainly CPO and PDO, both of which are constrained policy optimization approaches. The experimental results section requires more clarity and ease of presentation, as it is a bit difficult to follow what the results are trying to show exactly. However, the main conclusion is that PCPO significantly satisfies constraints in all the propoed benchmarks compared to the baselines. The authors compare to the sota baselines too for evaluating the significance of their approach.


Overall, I think the paper has useful merits - although it seems to be a computational ly challenging approach requiring second order approximations for both the KL terms (reward improvement and project step). It may be useful to see if there is a computationally simpler, PPO form of the variant that can be introduced for this proposed approach. I think it is useful to introduce such policy optimization methods satisfying constraints - and the authors in this work propose a simple approach based on projecting the policies into the constraint set, and solving the overall problem with convex optimization tools.  Experimental results are also evaluated with standard baselines, demonstrating the significance of the approach.


**Experience Assessment:**

I have published one or two papers in this area.

**Review Assessment: Checking Correctness Of Derivations And Theory:**

I assessed the sensibility of the derivations and theory.

**Review Assessment: Checking Correctness Of Experiments:**

I assessed the sensibility of the experiments.

**Review Assessment: Thoroughness In Paper Reading:**

I read the paper at least twice and used my best judgement in assessing the paper.

---

> ### Author Response · Authors · 2019-11-13
> **Response to Review #1**
>
> We thank Reviewer #1 for the helpful and insightful feedback. We have updated a version based on your suggestions. We provide answers to individual questions below.
>
> Reviewer #1’s comment #1: Comparison between theorem 3.1 and theorem 3.2, and the proof
> Response: The difference between these theorems is that theorem 3.1 deals with the constraint-satisfying case, and theorem 3.2 deals with the constraint-violating case. Both need to be considered.
> As noted, the proof is built upon Achiam et al., 2017, but it needs to be modified and elaborated to account for the extra projection step.
>
> Reviewer #1’s comment #2: Approximations and computational efficiency
> Response: We agree that our algorithm has two approximations, which adds computation. However, we use the following approach to reduce this computational cost while ensuring safe policy improvement.
> (1) We only compute the Fisher information matrix once. The reason we can do that is the step size is small. Hence, we can reuse the Fisher matrix of the reward improvement step also in the KL divergence projection step.
> The experimental results imply that this approach does not hurt the performance of PCPO. We have updated the paper in Section 6 to illustrate this approach.
>
> (2) We use the conjugate gradient method to avoid taking the inverse of the Fisher matrix. This further reduces the computational cost. The convergence of the conjugate gradient method is controlled by the condition number of the Fisher matrix: the larger the condition number, the more epochs needed to obtain an accurate approximation. Hence there is a tradeoff between the computational cost and the approximation accuracy. Please see Appendix F for an example of the approximation error and the computational cost of the conjugate gradient method.
>
> Reviewer #1’s comment #3: Presentation of the experimental results section
> Reponse: Thank you for your suggestion. We have updated the paper in Section 6.

---

### Official Review · AnonReviewer2 · 2019-10-23
**Official Blind Review #2**

**Rating:** 6

**Review:**

This paper proposes a new algorithm - Projection based Constrained Policy Optimization, that is able to learn policies with constraints, i.e., for CMDPs. The algorithm consists of two stages: first an unconstrained update for maximizing reward, and the second step for projecting the policy back to the constraint set.  The authors provide analysis in terms of bounds for reward improvement and constraint violation.  The authors characterize the convergence with two projection metrics: KL divergence and L2 norm.  The new algorithm is tested on four control tasks: two mujoco environments with safety constraints, and two traffic management tasks, where it outperforms the CPO and lagrangian based approaches.


This is an interesting work with impressive results.  However, this work still has a few components that need to be addressed and further clarification on novelty.  Given these clarifications in an author's response, I would be willing to increase the score.


1) Incremental work
The work extends the CPO [1] with a different update rule. Instead of having the update rule of CPO that does reward maximization and constraint satisfaction in the same step, the proposed update does that in two steps.  The theory and the algorithm stem directly from the original CPO work, including appendix A-C. The authors claim that another benefit of PCPO is that it requires no hyper-tuning, but same is true for CPO (in the sense that they both don’t need Lagrange multiplier) .


2) The utility of the performance bounds and fixed point
The performance bounds depend on the variable $\delta$, which is never explained. I’m assuming it is the same $\delta$ that is used in Lemma A.1. In that case, Theorem 4.1 tells about the existence of the fixed point of the algorithm under the assumptions specified Sec 4 (smooth objective function, twice differentiable, Hessian is positive definite).  There is no discussion regarding the comparison of the fixed-point of the algorithm with the optimal value function/policies. Also, all the analysis is with Hessian, whereas in the algorithm the Hessian is approximated via conjugate descent.


3) How is line-search eliminated?
One of the benefits of the proposed algorithm is that it doesn’t require line search (Sec 1). The underlying algorithm is still based on monotonic policy improvement theory in general, and more specifically on TRPO, so it should still have line-search as part of the optimization procedure.




References:

[1] Joshua Achiam, David Held, Aviv Tamar, and Pieter Abbeel. Constrained policy optimization. In Proceedings of International Conference on Machine Learning, pp. 22–31, 2017.


**Experience Assessment:**

I have published one or two papers in this area.

**Review Assessment: Checking Correctness Of Derivations And Theory:**

I assessed the sensibility of the derivations and theory.

**Review Assessment: Checking Correctness Of Experiments:**

I assessed the sensibility of the experiments.

**Review Assessment: Thoroughness In Paper Reading:**

I read the paper thoroughly.

---

> ### Author Response · Authors · 2019-11-13
> **Response to Review #2**
>
> We thank Reviewer #2 for the helpful and insightful feedback. We have updated a version based on your suggestions. We provide answers to individual questions below.
>
> Reviewer #2’s comment #1: Incremental work
> Response: We agree that our algorithm, PCPO, builds upon and extends the work of CPO. We identify that one issue of CPO is that its base optimization problem becomes infeasible when there is no intersection between the trust region and the cost constraint set. CPO did recovery by directly minimizing the cost constraint, which results in inefficient learning of the constraint-satisfying policy.
> Instead we adapt a two-step approach that combines a reward improvement step and a projection step to ensure efficient learning of the constraint-satisfying policy. The algorithm is motivated by a novel point of view of information geometry around the policy, and the property of the projection (shown in Appendix A and B).
> For the hyper-tuning comment, we apologize that in the final paragraph of Section 6, we actually compare PCPO with the Lagrange multiplier types of the variant, PDO and FPO. We have updated the paper.
>
> Reviewer #2’s comment #2: The utility of the performance bounds and fixed point
> Response: We apologize for the missing definition and the discussion. The variable $\delta$ is the step size for a reward improvement step. And it is the same in Theorem 3.1, Theorem 3.2, and Theorem 4.1. We have made this update.
> The fixed point of PCPO with KL divergence projection is the gradient of the objective. It belongs to the negative normal cone of the constraint set, i.e., the gradient of the cost constraint function. On the other hand, the fixed point of PCPO with L2 norm projection is the product of the inverse of the Fisher information matrix and the gradient of the objective. It belongs to the negative normal cone of the constraint set. Hence, these two projections converge to different fixed points. We have added discussion after Theorem 4.1. In Appendix G, we use an example to demonstrate that KL divergence and L2 norm projections have different optimization paths with convergence to different fixed points.
> The conjugate gradient method is to compute the product of the inverse of the Fisher and the gradient vector efficiently since the neural network policies are often large in practice. Hence, the analysis in Theorem 4.1 still holds since the algorithm still uses the information of H.
>
> Reviewer #2’s comment #3: How is line-search eliminated?
> Response: We do not use line search. Instead our algorithm reconciles the constraint violation (if any) by directly projecting the policy back onto the constraint set. This allows us to perform efficient updates in learning constraint-satisfying policies while not violating the constraints. We agree that we could use line search but it is not necessary.

---

### Official Review · AnonReviewer3 · 2019-10-23
**Official Blind Review #3**

**Rating:** 6

**Review:**

The paper proposes a technique to handle a certain type of constraints involved in Markov Decision Processes (MDP). The problem is well-motivated, and according to the authors, there is not much relevant work. The authors compare with the competing methods that they think are most appropriate. The numerical experiments seem to show superiority of their method in most of the cases. The proposed method has 4 main variants: (1) define projection in terms of Euclidean distance or (2) KL-divergence, and (a) solve the projection problem exactly (usually intractable) or (b) solve a Taylor-expanded variant (so there are variants 1a,1b,2a,2b).

Unfortunately, I do not feel well-qualified enough in the MDP literature to comment on the novelty, and appropriateness of comparisons. For now, I will take the authors' word, and rely on other reviewers.  The motivation of necessity of including constraints did seem persuasive to me.

Overall, this seems like a nice contribution based on the importance of the problem and the good experimental results, hence I lean toward accepting.  I do have some concerns that I mention below (chiefly that the theory presented is a bit of a red herring), but it may be that the overall novelty/contribution outweight these concerns:

(1) Concern 1: the theorems (Thm 3.1, 3.2) apply to the intractable version, and so are not relevant to the actual tractable version of the algorithm. These are nice motivations, but ultimately we're left with a heuristic method. Perhaps you can borrow ideas from the SQP literature?

(2) Concern 2: Fig 3(e), "Grid" data, your algo with KL projection does worse in Reward than TRPO, which is not unexpected since TRPO ignores constraints. But the lower plot shows that TRPO actually outperforms your KL projection-based algorithm even in terms of constraint!  By trying to respect the constraint, your algorithm has made things worse.  Can you explain this phenomenon?

(3) Concern 3: Thm 4.1 and Thm D.2, I don't know what you're proving because I don't know what f is. Please relate it to your problem and to your updates (e.g., Algo 1). If you are talking about just minimizing f(x) with convex quadratic constraints, then I think you are re-inventing the wheel (overall, your proof looks like you are re-inventing the wheel -- doesn't everything follow from the fact that projections operators are non-expansive?  If you scale with H (and assume it is positive definite) then you're still working in a Hilbert space, just with a non-Euclidean norm, and so you can re-use existing standard results on the convergence of projected gradient methods in Hilbert space to stationary points.


Smaller issues:

- The appendix was never referenced in the main paper. At the appropriate places in the main text, please mention that the proofs are in the appendix, and mention appendix C when discussing the PCPO update.

- For the PCPO update, the theorem needs to mention that H is positive definite. Using H as the Fisher Information matrix automatically guarantees it is positive semi-definite (please mention that), so the problem is at least convex, and then you assume it is invertible to get a unique solution.

- It wasn't obvious to me when the constraint set is closed and convex. Please discuss.

- When H is ill-conditioned, why not regularize with the identity? You switch between H and the identity, but why not make this more continuous, and look at H + mu I for small values of mu?

- Lemma D.1 is trivial if you use the definition of normal cones and subgradients. You also don't need to exclude theta from the set C, since if it is in the set C, then the quadratic term will be zero, hence less than/equal to zero.

**Experience Assessment:**

I have read many papers in this area.

**Review Assessment: Checking Correctness Of Derivations And Theory:**

I assessed the sensibility of the derivations and theory.

**Review Assessment: Checking Correctness Of Experiments:**

I did not assess the experiments.

**Review Assessment: Thoroughness In Paper Reading:**

I read the paper at least twice and used my best judgement in assessing the paper.

---

> ### Author Response · Authors · 2019-11-13
> **Response to Review #3**
>
> We thank Reviewer #3 for the helpful and insightful feedback. We have updated a version based on your suggestions. We provide answers to individual questions below.
>
> Reviewer #3’s concern #1: SQP for tractable version
> Response: When the step size $\delta$ is small, the tractable version of the algorithm is approximately closed to the intractable version. This indicates that Theorem 3.1 and Theorem 3.2 still provide intuition on the change of the reward and the constraint value.
> For the SQP suggestion, we have used the convergence analysis from sequential quadratic programing (SQP), and Theorem 4.1 is based on SQP to provide analysis for (1) stationary points of KL divergence and L2 norm projections, and (2) conditions about how the reward objective value (in the negative form) changes.
>
> Reviewer #3’s concern #2: Grid task
> Response: Thank you for pointing this out. We did not include the explanation for this point in the original paper.
> The reason for this observation is that the Fisher matrix is ill-conditioned. As a result, it gives an inaccurate answer. This inaccurate approximation leads to more constraint violation than TRPO. This is not a problem in the other examples. We include this negative result to illustrate the algorithm is not universal, and does depend on the application.
> In Appendix F, we point out that there is a tradeoff between the approximation error and the computational cost of the conjugate gradient method. We show that by having more iterations resolves the above problem. However, it increases the computational cost.
>
> Reviewer #3’s concern #3: Analysis in Theorem 4.1
> Response: We apologize that we did not define clearly what $f$ is in the original paper. The $f$ in Theorem 4.1 is the negative reward objective function $J^{R}$. We minimize $f$ to follow the convention of the literature that authors typically minimize the objective function. We have updated the paper to make it clear in Theorem 4.1.
> We acknowledge that the proof of Theorem 4.1 is based on working in a Hilbert space and the non-expansive property of projection. However, Theorem 4.1 is important since it provides a theoretical understanding for the observation we make in the experimental results as well as characterizing the difference between KL divergence and L2 norm projections. Theorem 4.1 is also useful in practice since it provides a prescription for choosing the type of projection depending on the problem. We have modified the proof of Theorem 4.1 in Appendix D, and mentioned that the proof is based on working in a Hilbert space and the property of projection.
>
> Smaller Issues:
> #1: Mention that the proofs are in Appendix.
> Response: Thank you for pointing this out. We have updated the paper.
>
> #2: Mention that H is positive semi-definite
> Response: Thank you for pointing this out. We have updated the paper.
>
> #3: Closeness and convexity of the constraint set
> Response: We apologize for the confusion. For a neural network policy, the constraint defined in Eq. (3) is usually non-convex, and hence the projection onto the non-convex set is not unique. However, in the tractable case (use approximation), the projection is unique since the constraint set is approximated by a convex half space $C=\{x:a^Tx\leq b\}$ with small step size $\delta$. We have removed this statement in the paragraph of the projection step in Section 3 to avoid confusion and for clarity of the presentation.
>
> #4: Ill-conditioned H
> Response: Thank you for the suggestion. We will try this out and leave it as future work for developing more continuous approach.
>
> #5: Lemma D.1
> Response: We include Lemma D.1 for completeness and the proof of Theorem D.2. As you suggest, we have updated Lemma D.1.

---

### Author Response · Authors · 2019-11-13
**General Reponse to All Reviewers**

We thank all the reviewers for the valuable feedback and constructive suggestions. We have updated a version of our paper, and provided clarification.

---

### Decision · Program_Chairs · 2019-12-19

**Decision:**

Accept (Poster)

**Comment:**

The paper proposes a new algorithm for solving constrained MDPs called Projection Based Constrained Policy Optimization. Compared to CPO, it projects the solution back to the feasible region after each step, which results in improvements on some of the tasks considered.

The problem addressed is relevant, as many tasks could have important constraints e.g. concerning fairness or safety.

The method is supported through theory and empirical results. It is great to have theoretical bounds on the policy improvement and constraint violation of the algorithm, although they only apply to the intractable version of the algorithm (another approximate algorithm is proposed that is used in practice). The experimental evidence is a bit mixed, with the best of the proposed projections (based on the KL approach) sometimes beating CPO but also sometimes being beaten by it, both on the obtained reward and on constraint satisfaction.

The method only considers a single constraint. I'm not sure how trivial it would be to add more than one constraint. The reviewers also mention that the paper does not implement TRPO as in the original paper, as in the original paper the step size in the direction of the natural gradient is refined with a line search if the original step size (calculated using the quadratic expansion of the expected KL) does violate the original constraints. (Line search on the constraint as mentioned by the authors would be a different issue). Futhermore, the quadratic expansion of the KL is symmetric around the current policy in parameter space. This means that starting from a feasible solution the trust region should always overlap with the constraint set when feasibility is maintained, going somewhat agains the argument for PCPO as opposed to CPO brought up by the authors in the discussion with R2. I would also show this symmetry in illustrations such as Fig 1 to aid understanding.